# Carbonate system distribution, anthropogenic carbon and acidification in the Western Tropical South Pacific (OUTPACE 2015 transect)

Thibaut Wagener[1], Nicolas Metzl[2], Mathieu Caffin[1], Jonathan Fin[2], Sandra Helias Nunige[1], Dominique Lefevre[1], Claire Lo Monaco[2], Gilles Rougier[1] and Thierry Moutin[1]

[1]Aix Marseille Univ, CNRS, IRD , Université de Toulon, Mediterranean Institute of Oceanography (MIO), UM 110 , 13288, Marseille, France
[2]Sorbonne Université, CNRS, IRD, MNHN, Laboratoire d'océanographie et du climat : expérimentation et approches numériques (LOCEAN), Case 100,  4 place Jussieu, 75252 Paris cedex 05, France.

*Correspondence to*: Thibaut Wagener (thibaut.wagener@univ-amu.fr)

**Abstract.**

The western tropical South Pacific was sampled along a longitudinal 4000 km transect (OUTPACE cruise, 18 Feb., 3 Apr. 2015) for measurement of carbonate parameters (total alkalinity and total inorganic carbon) between the Melanesian Archipelago (MA) and the western part of the South Pacific gyre (WGY). This manuscript reports this new dataset and derived properties: pH on the total scale ($pH_T$) and the $CaCO_3$ saturation state with respect to calcite ($\Omega_{cal}$) and aragonite ($\Omega_{ara}$). We also estimate anthropogenic carbon ($C_{ANT}$) distribution in the water column using the TrOCA method (Tracer combining Oxygen, inorganic Carbon and total Alkalinity). Along the OUTPACE transect a deeper penetration of $C_{ANT}$ in the intermediate waters was observed in the MA, whereas highest $C_{ANT}$ concentrations were detected in the sub-surface waters of the WGY. By combining our OUTPACE dataset with data available in GLODAPv2 (1974-2009), temporal changes in oceanic inorganic carbon were evaluated. An increase of 1.3 to 1.6 µmol kg$^{-1}$ a$^{-1}$ for total inorganic carbon in the upper thermocline waters is estimated whereas $C_{ANT}$ increases  1.1 to 1.2 µmol kg$^{-1}$ a$^{-1}$. In the MA intermediate waters (27 kg m$^{-3}$ < $\sigma_\theta$ < 27.2 kg m$^{-3}$) an increase of 0.4 µmol kg$^{-1}$ a$^{-1}$  $C_{ANT}$ is detected. Our results suggest a clear progression of ocean acidification in the western tropical South Pacific with a decrease of the oceanic $pH_T$ of up to -0.0027 a$^{-1}$ and a shoaling of the saturation depth for aragonite of up to 200 m since the pre-industrial period.

## 1    Introduction

Human activities inject about 10 PgC kg of carbon per year to the atmosphere which might have major consequences on climate. It is recognized that the ocean plays a key role in the control of atmospheric $CO_2$ through uptake by the so called "oceanic carbon pump". Through this "pump", the ocean sequesters ca. 25% of the $CO_2$ injected annually in the atmosphere by human activities (Le Quéré et al., 2018). A consequence of the ocean carbon uptake is a decrease of the oceanic pH (Feely et al, 2004) which is described as ocean acidification (the so-called "other" $CO_2$ problem). Effects of ocean acidification

have been observed on marine organisms and could affect the marine ecosystems (Riebesell et al., 2000). Improving our understanding of the oceanic $CO_2$ uptake relies primarily on observations of the marine carbonate cycle. Studies on the oceanic carbonate cycle have been mostly conducted in the frame of international programs. The World Ocean Circulation Experiment (WOCE) and the Joint Global Flux Study (JGOFS) in the 90's have coordinated oceanographic cruises along large sections in the ocean to collect samples through the water column and to perform accurate measurements of carbonate parameters and ancillary parameters (temperature, salinity, dissolved oxygen, nutrients,...). Since 2000, efforts have been made to "revisit" oceanic sections according to the WOCE strategy in order to assess oceanic changes at the scale of a decade. These programs have generated important databases for oceanic carbonate chemistry (e.g. GLODAP_V2 – Olsen et al., 2016).

In order to better assess the role of the ocean on the global carbon cycle, the concept of oceanic anthropogenic carbon ( $C_{ANT}$) has been introduced and refers to the fraction of dissolved inorganic carbon ($C_T$) in the ocean that originates from carbon injected by human activities in the atmosphere since the industrial revolution. As $C_{ANT}$ is not a directly measurable quantity, it can only be estimated through assumptions that are subjected to intense scientific debate (Sabine and Tanhua, 2010). In particular, it has been recently recognized that ocean circulation changes drive significant variability in carbon uptake (De Vries et al., 2017). Detecting, separating and attributing decadal changes of the carbonates system ($C_T$ and $A_T$), $C_{ANT}$ and pH in the ocean at global or regional scales remains challenging.

Within this context, the Pacific Ocean is a particularly challenging area to study due to its size (ca. one third of the Earth's and one half of the oceanic surface). Even if, due to its remoteness from land, it remains largely under-explored by oceanographic vessels compared to other oceanic areas, the Pacific Ocean has been covered by cruises along long sections (the "P sections" from the WOCE program). Most of these sections have been revisited during the last years (see for example Sabine et al., 2008 or Kouketsu et al., 2013). In a recent study based on repeated sections in the Pacific (P16 at 150W), Carter et al (2017) observed significant increase of Cant in the top 500m around 10°S-30°S and a local carbon storage maximum around 20°S in recent years (between 2005 and 2014). In this context, the OUTPACE data presented in this study, associated to historical observations (since the pioneer 1974 GEOSECS) offer a new view to evaluate variability and decadal changes of $C_T$, $C_{ANT}$ and pH in the tropical Pacific, here focused in the western tropical south pacific (WTSP).

The aim of this paper is to report a new dataset of oceanic inorganic carbon (based on measurements of $C_T$ and total alkalinity ($A_T$) ) acquired in the WTSP during the OUTPACE (Oligotrophic to UltTra oligotrophic PACific Experiment) cruise performed in 2015 (Moutin et al., 2017). The main focus of the OUTPACE cruise was to study the complex interactions between planktonic organisms and the cycle of biogenic elements on different scales, motivated by the fact that the WTSP has been identified as a hot spot of $N_2$ fixation (Bonnet et al., 2017). The data presented here have been partially used in another paper of the special issue (Moutin et al., 2018) in order to study the biological carbon pump in the upper (surface to 200m) water column. In this paper we will explore the carbonate data between the surface and 2000 m depth. The OUTPACE transect (Figure 1) is close to existing WOCE and GO-SHIP lines in the South Pacific : it is parallel to the zonal P21 line (18° S visited in 1994 and 2009) and the P06 line (32° S visited in 1992, 2003 and 2010), it is crossed by the

meridional P14 line (180° E visited in 1994 and 2007) and P15 line (170° W visited in 2001, 2009 and 2016) and it is situated at the eastern side of the P16 line (150° W visited in 1992, 2005 and 2014). However, the OUTPACE transect is not corresponding to any earlier occupation of the "WOCE lines" in the South Pacific and no tracers of water mass age were measured during the cruise, which limits the possibilities of a robust analysis of $C_{ANT}$ accumulation in the area. Moreover, the horizontal and vertical resolution of the OUTPACE dataset is low. In consequence, the OUTPACE dataset can not be used to look at decadal changes in $C_{ANT}$ content in the South Pacific (e.g. Carter et al. 2017, Kouketsu et al. 2013). Here, $C_{ANT}$ estimates based on the TrOCA (Tracer combining Oxygen, inorganic Carbon and total Alkalinity) method will be used as as a tool to investigate changes in $C_T$. Moreover, by comparing our data with the high quality data (internally consistent through a secondary quality control (Olsen et al., 2016)) available in the Global Ocean Data analysis Project version 2 (GLODAPv2 database), will allow to evaluate $C_T$, $A_T$, $C_{ANT}$ (for TrOCA) and $pH_T$ (pH on total scale) trends in sub surface waters and at depth.

The paper is organized as follows: After describing the methods used to acquire the dataset and the way the auxiliary data have been used in Sect. 2, we briefly present the hydrographic context of the cruise in Sect. 3. We then present in Sect. 4, the carbonate dataset acquired during the cruise. In Sect. 5, estimated $C_{ANT}$ values in the water column are presented, the validity of these estimates based on the TrOCA method is discussed and geographical patterns are evoked. In Sect. 6, the temporal changes in oceanic inorganic carbon in the WTSP combining data available in GLODAPv2 and our OUTPACE dataset are presented and discussed. Finally, in Sect. 7, some features in relation to ocean acidification are inferred from our dataset.

## 2    Material and Methods

### 2.1    Cruise and sampling strategy

The OUTPACE cruise took place between 18 February and 3 April 2015 from Noumea (New Caledonia) to Papeete (French Polynesia), in the WTSP on board the French research vessel "L'Atalante" (Fig. 1). A total of 18 stations were sampled mostly in the top 2000 m of the water column along a ~4000 km transect from the Melanesian archipelago to the South Pacific gyre (Moutin et al., 2017). A CTD-Rosette was deployed to acquire data with CTD and associated sensors along vertical profiles and to collect discrete seawater samples from 24 12-L Niskin bottles for chemical analysis. Due to technical failures on the main CTD-Rosette, for two of the casts considered in this study, a trace metal clean CTD rosette (TM-R) equipped with 24 teflon-lined GO-FLO bottles devoted to trace metal analyses was used. The configurations of both CTD Rosettes are detailed elsewhere (Moutin et al., 2017).

For carbonate parameters, seawater was sampled from 31 casts over the 18 stations. At each station, on a regular basis, samples were collected at 12 depths between the surface and 2000 m on two distinct casts: 6 samples on a 0-200 m cast and 6 samples on a 0-2000 m cast. At station SD-13, only one cast was sampled down to 500 m depth. In addition, at station LD-C, samples were collected at 24 depths on a deep cast (down to 5000 m) and 12 samples were collected at the same depth (25 m) on a "repeatability" cast. Details on the casts performed for this study are summarized in Table 1.

## 2.2 Chemical measurements on discrete samples

All samples were collected within less than 1 hour after arrival of the CTD rosette on deck.

### 2.2.1 Total alkalinity and dissolved inorganic carbon

Samples for $A_T$ and $C_T$ were collected in one 500 mL borosilicate glass flask (Schott Duran®) and poisoned immediately after collection with $HgCl_2$ (final concentration 20 mg.L$^{-1}$). Samples were stored at 4°C during transport and were analyzed (within 10 days) 5 months after the end of the cruise at the SNAPO-CO$_2$ (Service National d'Analyse des paramètres Océaniques du CO$_2$- LOCEAN – Paris). $A_T$ and $C_T$ were measured on the same sample based on a potentiometric titration in a closed-cell (Edmond, 1970). A non-linear curve fitting approach was used to estimate $A_T$ and $C_T$ from the recorded titration

data (Dickson 1981, DOE 1994). Measurements were calibrated with Certified Reference Material (CRM) provided by Dr. A Dickson, Univ. Southern California (Batch 139 - $C_T$ : 2023.23 ± 0.70 µmol kg$^{-1}$ and $A_T$: 2250.82 ± 0.60 µmol kg$^{-1}$, see Dickson, 2010). The reproducibility, expressed as the standard deviation of the CRM analysis (n=15), was 4.6 µmol kg$^{-1}$ for $A_T$ and 4.7 µmol kg$^{-1}$ for $C_T$ . Based on replicate measurements at station LD-C (cast out_c_194, see Table 1) the reproducibility, expressed as the standard deviation of the analysis of the replicates collected at the same depth (ca. 25 m,

n=12) from different Niskin bottles was 3.6 µmol kg$^{-1}$ for $A_T$ (average value = 2324.7 µmol kg$^{-1}$ ) and 3.7 µmol kg$^{-1}$ for $C_T$ (average value = 1969.7 µmol kg$^{-1}$ ).

### 2.2.2 Oxygen concentration

Dissolved oxygen concentration [$O_2$] was measured following the Winkler method (Winkler, 1888) with potentiometric endpoint detection (Oudot et al., 1988). For sampling, reagents preparation and analysis, the recommendations from

115 Langdon (2010) were carefully followed. The thiosulfate solution was calibrated by titrating it against a potassium iodate certified standard solution of 0.0100N (CSK standard solution – *WAKO*). The reproducibility, expressed as the standard deviation of replicates samples was 0.8 µmol kg$^{-1}$ (n=15, average value= 195.4 µmol kg$^{-1}$).

## 2.3 Vertical profiles of hydrological and biogeochemical parameters

### 2.3.1 CTD measurements

CTD measurements were ensured by a Seabird™ 911+ underwater unit which interfaced an internal pressure sensor, two redundant external temperature probe (SBE3plus) and two redundant external conductivity cells (SBE4C). The sensors were calibrated pre- and post-cruise by the manufacturer. No significant drift between the redundant sensors was observed. For vertical profiles, full resolution data (24 Hz) were reduced to 1 dbar binned vertical profiles on the downcast with a suite of processing modules using the Seabird™ dedicated software (*SbeDataProcessing*). For values at the closure of the Niskin

bottles, values collected at 24 Hz were averaged 3 s before and 5 s after closure of the bottle. In this study, for temperature

and conductivity the signal of the first sensors has been systemically used. For the two TM-R casts, no significant difference with the main CTD-Rosette on temperature and conductivity was observed.

### 2.3.2 Oxygen measurements

$[O_2]$ was also measured with a SBE43 electrochemical sensor interfaced with the CTD unit. The raw voltage was converted to oxygen concentration with 13 calibration coefficients based on the Seabird™ methodology derived from Owens and Millard (1985). Three of these coefficients (the oxygen signal slope, the voltage at zero oxygen signal, the pressure correction factor) were adjusted with the concentrations estimated with the Winkler method on samples collected at the closure of the bottles. One unique set of calibration coefficients has been applied to all oxygen profiles from the cruise because no significant drift of the sensor was observed during the time of the cruise. For the two TM-R casts, values have been corrected with a drift and offset based on the comparison of 15 pairs of casts (main CTD-rosette / TM-R) collected close in time (less than 2 h) and space (less than 1 nautical mile) over the entire OUTPACE transect.

### 2.4 Derived parameters

Practical salinity ($S_P$) was derived from conductivity, temperature and pressure with the EPS-78 algorithm. Absolute salinity ($S_A$), potential temperature ($\theta$), conservative temperature ($\Theta$) and potential density ($\sigma_\theta$) were derived from Sp, temperature, pressure and the geographic position with the TEOS-10 algorithms (Valdarez et al., 2011). These five derived parameters were calculated within the processing with *Seabird*™ dedicated software.

Seawater pH on the total scale ($pH_T$) and the $CaCO_3$ saturation state with respect to aragonite ($\Omega_{ara}$) were derived from $A_T$ and $C_T$ with the "Seacarb" R package (Gattuso and Lavigne, 2009). $CaCO_3$ saturation state with respect to calcite was not considered because seawater up to 2000 dbar was supersaturated with respect to calcite ($\Omega_{cal} > 1$). Following the recommendations from Dickson et al. (2007), the constants for carbonic acid $K_1$ and $K_2$ from Lueker et al. (2000), the constant for hydrogen fluoride $K_F$ from Perez and Fraga (1987) and the constant for hydrogen sulfate $K_S$ from Dickson (1990) were used. Orthophosphate and silicate concentration were considered in the calculation. Methods for nutrients measurement are presented in details in Fumenia et al. (2018). When nutrient data were not available (Station SD-8), silicate and orthophosphate were estimated from the nutrient profile measured on cast out_c_163 (interpolated values). Apparent Oxygen Utilization (AOU) was computed from the difference between oxygen solubility (at p=0 dbar, $\theta$ and $S_P$) estimated with the "Benson and Krause coefficients" in Garcia and Gordon (1992) and in situ $[O_2]$.

For estimation of $C_{ANT}$, the TrOCA method was used. The TrOCA approach was first proposed in Touratier and Goyet (2004a, b) with improvements in Touratier et al. (2007). In brief, the TrOCA parameter is defined as a combination of $A_T$, $C_T$ and $[O_2]$ that accounts for biologically induced relative changes among these parameters (with constant stoichiometric ratios). TrOCA is thus a quasi-conservative tracer derived from $C_T$ in the ocean. Within a defined water mass, changes in TrOCA over time are independent from biology and can be attributed to the penetration of $C_{ANT}$. In consequence $C_{ANT}$ can be calculated in a parcel of water from the difference between current and pre-industrial TrOCA (TrOCA°) divided by a

stoichiometric coefficient. The simplicity of the TrOCA method relies on the fact that a simple formulation for TROCA° has been proposed based on potential temperature and alkalinity and thus an estimation of $C_{ANT}$ can be done by a simple

calculation using $C_T$, $A_T$, $[O_2]$ and $\theta$. In this study, the formulation proposed in eq. 11 in Touratier et al. (2007) is used to calculate $C_{ANT}$ and is reminded here in Eq. (1).

$$C_{ANT} = \frac{\left[O_2\right] + 1.279\left(C_T - \dfrac{A_T}{2}\right) - \exp\left(7.511 - 1.087 . 10^{-2}\theta - \dfrac{7.81 . 10^{-5}}{A_T^2}\right)}{1.279} \quad (1)$$

This formulation is based on an adjustment of the TrOCA coefficients using $\Delta^{14}C$ and CFC-11 from the GLODAP-V1 database (Key et al., 2014). Touratier et al. (2007) estimated the overall uncertainty of the $C_{ANT}$ with TrOCA method to c.a. 6

µmol kg$^{-1}$ based on the random propagation of the uncertainties on the variables ($C_T$, $A_T$, $[O_2]$ and $\theta$) and coefficients used in Eq. 1. The limitations and validity of the TrOCA method will be discussed in details in Sect. 5.

## 2.5 Data from available databases

For comparison with existing values of carbonate chemistry in the area of the OUTPACE cruise, relevant data were extracted from GLODAPv2 database (NDP-93 - Olsen et al. 2016). The specific data file for the Pacific Ocean was used (downloaded

from https://www.nodc.noaa.gov/ocads/oceans/GLODAPv2/ on December 14, 2017). For comparison with OUTPACE data, GLODAPv2 data were selected between 22°S and 17°S and between 159°E and 159° W (going westwards). For specific comparisons in the Melanesian archipelago (MA) and the South Pacific Western Gyre waters (WGY) a zonal subset of the extracted data was used: 159°E and 178° W for MA and 170°W to 159°W for WGY (see Fig. 1).

## 3 Hydrological context along the OUTPACE transect

The hydrological context encountered during the OUTPACE transect is presented with a $\Theta$ – $S_A$ diagram between 0 and 2000 dbar on Fig. 2. A detailed description of the water masses encountered during the OUTPACE cruise can be found in Fumenia et al. (2018). Briefly, from the surface to 2000 dbar, the following features are distinguished: the surface waters ($\sigma_\theta < 23.5$) were characterized by temperatures over 25 °C with increasing temperature and salinity towards the east and AOU close to zero. Under the the surface water, the upper thermocline waters (UTW) presented a maximum in salinity reaching values

higher than 36 g kg$^{-1}$ in the eastern part of the cruise. In the lower thermocline waters, $S_A$ decreased with depth with a more pronounced decrease in the eastern part than in the western part whereas AOU is higher in the eastern part than in the western part of the studied area. These differences in lower thermocline waters have been described for South Pacific Central Waters (SPCW) with more saline western (WSPCW) and less saline eastern (ESPCW) waters (Tomczack and Godfrey, 2001). Below the thermocline, intermediate waters are constituted of Sub-Antarctic Mode Waters (SAMW) and Antarctic

Intermediate Waters (AAIW). AAIW have a salinity minimum close to the $\sigma_\theta = 27$ kg m$^{-3}$ isopycnal . Hartin et al. (2011) defines SAMW with $\sigma_\theta$ values between 26.80 and 27.06 kg m$^{-3}$ corresponding to a minimum in potential vorticity, and AAIW

with $\sigma_\theta$ values between 27.06 and 27.40 kg m$^{-3}$. The separation of both waters is not trivial in the subtropical area. SAMW is generally associated to lower AOU than AAIW. Finally deep waters constituted of Upper Circumpolar Deep Waters (UCDW) correspond to an increase in salinity and AOU for depth corresponding to $\sigma_\theta > 27.4$ kg m$^{-3}$.

In this study, discussion will sometimes make distinction between two sub-regions along the OUTPACE transect: MA and WGY (See Sect. 2.5 for definition). This distinction is mainly based on geographic and oceanographic arguments. Indeed, these two sub-regions are geographically separated by the Tonga volcanic arc. WGY is characterized by higher surface temperature and a higher salinity in the upper thermocline waters than MA. The difference between these sub-regions is evidenced by the difference in oligotrophy (Moutin et al., 2018). Due to specific conditions in the transition area between the

MA and WGY (de Verneil et al., 2017), SD11, SD12 and LDB were discarded from both groups in this study following the arguments in Moutin et al. (2018).

## 4    Carbonate chemistry along the OUTPACE transect

$A_T$ and $C_T$ measured along the OUTPACE transect are presented on Fig 3a and 3b. All vertical profiles for $A_T$, $A_T$ normalized to $S_A = 35$ g kg$^{-1}$ ($A_{Tn35}$) and $C_T$ are presented on Fig. 3e, 3f and 3G. $A_T$ ranged between 2300 and 2400 µmol.kg$^{-1}$.

Below the surface, a pronounced maximum in $A_T$ was observed associated to the saltier upper thermocline waters. When normalized to $S_A = 35$ g kg$^{-1}$, $A_T$ values are remarkably constant in the upper 500 dbar with values between 2270 and 2310 µmol kg$^{-1}$. Below 500 dbar, $A_T$ increases with depth up to ca. 2400 µmol kg$^{-1}$ indicating that alkalinity changes are mostly due to salinity changes in the upper water column whereas the increase in the deep waters is mainly due to carbonate biominerals remineralization. $C_T$ values are close to 1950 µmol kg$^{-1}$ in the surface and increase with depth up to 2300 µmol

kg$^{-1}$ at 2000 dbar. The $C_T$ gradient in the upper water column has been described in Moutin et al. (2008). Below 2000 dbar, $C_T$ is relatively invariant with slightly lower values in the bottom waters (below 4000 dbar)due to the presence of very old deep waters originating from the north Pacific relative to the northward moving bottom waters that have not accumulated as much carbon (Murata et al. 2007). $A_T$ and $C_T$ values in deep waters measured during OUTPACE are in good agreement with the data of the GLODAPv2 database (Fig. 3E, 3f et 3g). No systematic adjustment of the OUTPACE dataset with the

GLODAPv2 dataset was performed because only very few data are available in the deep ocean where crossover comparison can be performed for cruises carried out in different decades. Nevertheless, for the only "deep" cast performed during OUTPACE (out_c_163 at station C), we performed a simple crossover analysis with the station 189 (located at 107 km kilometers from OUTPACE station C) of the Japanese "P21 revisited" cruise in 2009. We compared interpolated profiles on density surfaces values ($27.75 < \sigma_\theta < 27.83$ corresponding to pressure levels of ca. 3000 to 5500 dbar) . The estimated offsets

are -2.0 ± 4.2 µmol kg$^{-1}$ for $A_T$ and -2.0 ± 4.4 µmol kg$^{-1}$ for $C_T$ suggesting measurement biases are likely no larger. This simple quality control procedure seems to indicate that no systematic adjustment is needed.

Derived parameters from the $A_T$ and $C_T$ measurements are presented on Fig. 3c for $pH_T$ values (estimated at in situ temperature and pressure). $pH_T$ decreases from values close to 8.06 in surface to values close to 7.84 at 2000 m. Surface

values of $pH_T$ are typical of subtropical warm waters and are in a similar range as the austral summer values estimated by Takahashi et al. (2014) in this area. (8.06 - 8.08). Figure 3d represents the vertical distribution of computed values of $\Omega_{ara}$ along the OUTPACE transect. Seawater is supersaturated with respect to aragonite ($\Omega_{ara}>1$) at surface with $\Omega_{ara}$ values of ca. 4.0 again in good agreement with the austral summer values of 4 – 4.4 estimated by Takahashi et al. (2014) in this area. Values of $\Omega_{ara}$ decrease with depth and seawater becomes undersaturated with respect to aragonite ($\Omega_{ara}<1$) at an horizon situated below 1000 dbar in the west and above 1000 dbar in the eastern part of the cruise, with a general shoaling of the $\Omega_{ara}$ values from west to east, in good agreement with a previous study by Murata et al. (2015) in this area.

## 5    Anthropogenic carbon estimation along the OUTPACE transect

The TrOCA method is a way to quantify $C_{ANT}$ in the ocean based on $C_T$, $A_T$, $[O_2]$ and $\theta$. This method has been used and compared to other methods in different oceanic areas (e.g. Lo Monaco et al., 2005; Alvarez et al., 2009; Vazquez-Rodriguez et al.,2009) : based on specific $C_{ANT}$ inventories in the water column, the TrOCA method reasonably agreed with the other methods (including transient tracer based method). However, Yool et al. (2010) "tested" the TrOCA method within an ocean general circulation model and argued that the use of globally uniform parametrization for the estimation of the preindustrial TrOCA is a source of significant overestimation but also that even with regionally "tunned" parameters a global positive bias in the method exists. As no tracers of water mass age were measured during the OUTPACE cruise, the main motivation for using the TrOCA method was to make $C_{ANT}$ estimations based on a simple calculation from parameters acquired within the cruise as done in other cruises conducted in south tropical Pacific waters (e.g. Azouzi et al., 2009; Ganachaud et al., 2017). Even if $C_{ANT}$ estimates from TrOCA could be biased, the application of a simple back-calculation method that accounts for biologically induced relative changes in $C_T$ is used here to identify some spatial features in the distribution of the carbonate system along the OUTPACE transect. Based on Yool et al. (2010), the error on the TrOCA $C_{ANT}$ estimates will be considered here as to the normalized standard deviation of 1,67 for the TrOCA variant optimized with world ocean data (See Table 2 in Yool et al. 2010).

As mentioned by Touratier et al. (2007), $C_{ANT}$ estimates cannot be considered within the mixed layer because the underlying hypotheses used in the formulation of TrOCA may not be verified due to biological activity and gas transfers across the air–sea interface. To avoid this issue, $C_{ANT}$ estimates are generally used below the "permanent" mixed layer depth (e.g. Alvarez et al., 2009, Carter et al., 2017). For the OUTPACE area, Moutin et al. (2018) shows that the mixed layer depth do not exceed 70 m in the area. Even if the depths of the deep chlorophyll maximum was encountered below 100 dbar along the transect, we will consider $C_{ANT}$ values up to 100 dbar. It can be mentioned that the $C_{ANT}$ values of 50-60  $\mu mol\ kg^{-1}$ in the top of the water column (100 dbar), are in reasonable agreement with a rough estimate of thermodynamic consistent $C_T$ changes: by assuming that $CO_2$ in surface seawater is in equilibrium with the atmosphere, we estimated that with a partial pressure of $CO_2$ ($pCO_2$) of 280 µatm at the pre-industrial period, a $pCO_2$ of 380 µatm during OUTPACE (Moutin et al., 2018) and a constant $A_T$ over time of 2300 µmol $kg^{-1}$, $C_T$ change in surface waters between pre-industrial and 2015 is of ca. 65 µmol kg-1

for a temperature of surface waters between 25 and 28 °C. For OUTPACE, $C_{ANT}$ estimates below 1000 dbar, were not significantly different from 0 µmol kg$^{-1}$ with a standard deviation of 6.3 µmol kg$^{-1}$.

$C_{ANT}$ distribution along the OUTPACE transect is presented on Fig. 4a and all vertical profiles for $C_{ANT}$ are presented on Fig. 4b with a more detailed view of the first 1500 dbar of the water column on Fig. 4c. Figures 4b and 4c distinguish values from

the MA and the WGY area. The $C_{ANT}$ vertical profiles suggest a penetration of anthropogenic carbon up to 1000 dbar. As mentioned before, estimated values of $C_{ANT}$ reach values of 60 ± 40 µmol kg$^{-1}$ at depth of 100 dbar, then regularly decreases to values close to 10 - 20 ± 13 µmol kg$^{-1}$ at a depth of 1000 dbar and reaches values close to 0 µmol kg$^{-1}$ below 1500 dbar. The zonal $C_{ANT}$ section along the OUTPACE transect (Fig. 4a) presents two features: (1) a deeper penetration of $C_{ANT}$ in the western part of the transect with values of $C_{ANT}$ reaching 40 µmol kg$^{-1}$ around the isopycnal layer of 27 kg m$^{-3}$ (ca. 700 dbar)

with a coherent behaviour with the distribution of AOU and (2) a larger accumulation of $C_{ANT}$ in the eastern part of the transect centred around the isopycnal layer of 25 kg m$^{-3}$ (ca. 200 dbar).

Several studies have identified deeper $C_{ANT}$ penetration in the Western South Pacific than in the Eastern South Pacific at tropical and subtropical latitudes. The primary reason for this longitudinal difference might be associated to deeper convection in the western part and upwelling in the eastern part. AAIW has been described as the lower limit of the

penetration of $C_{ANT}$ in the ocean interior of the South Pacific (Sabine et al., 2004). Moreover, a recent study by deVries et al., (2017) shows that ocean circulation variability is the primary driver for changes in oceanic $CO_2$ uptake at decadal scales. Based on $C_T$ changes between the two repeated visits of the longitudinal P21 line (18° S close to the OUTPACE transect) in 1994 and 2009, Kouketsu et al. (2013) shows faster increase of $C_{ANT}$ in the western part than in the eastern part of the section. They also postulate that $C_{ANT}$ may have been transported by deep circulation associated to the AAIW. In the subtropical

Pacific along the P06 line (longitudinal section at ca. 32°S), Murata et al. (2007), also identified an increase of $C_{ANT}$ in the SAMW and AAIW. Waters et al. (2011), based on the extended multiple linear regression (eMLR) method along the P06 line (and taking into account a third visit) attributes the deeper penetration of $C_{ANT}$ in the western part of the section to the local formation of subtropical mode water in the area.

In the eastern part of the OUTPACE cruise, the detected accumulation of $C_{ANT}$ in the upper thermocline waters may be

related to recent observations of a significant accumulation of $C_{ANT}$ at latitudes around 20°S on the P16 meridional transect along 150°W by Carter et al. (2017). This change in $C_{ANT}$ accumulation is attributed to changes in the degree of the water mass ventilation due to variability in southern Pacific subtropical cell. Along the P16 line, Carter et al. (2017) observed high values of $C_{ANT}$ (up to 60 µmol.kg$^{-1}$) for the upper water column at the latitude of OUTPACE area in good agreement with our estimates in WGY in the upper water column. Finally, it should also be mentioned that, due to the presence of one of the

main OMZ area, denitrification occurs in the eastern South Pacific and can be traced by the N* parameter (Gruber and Sarminento, 2007). Denitrification, by transforming organic carbon to inorganic carbon without consumption of oxygen, could induce an overestimation of $C_{ANT}$ by the TrOCA method (and other back calculation methods) due to a biological release of $C_T$ that is not taken into account in the formulation of the quasi conservative TrOCA tracer. Horizontal advection by the south equatorial current of the strong negative N* signal originating from the Eastern Pacific towards the western

Pacific was previously described (Yoshikawa et al., 2015). Fumenia et al. (2018) have estimated N* along the OUTPACE transect and show slightly negative N* values in the upper thermocline waters at the eastern side of the OUTPACE transect where the highest $C_{ANT}$ values are estimated. However, Murata et al. (2007) showed that, based on a direct relation between $C_T$ and N*, the influence of denitrification should be negligible on $C_{ANT}$ estimations in this area. Therefore, the N* correction has not been introduced in the $C_{ANT}$ estimates and the effect of denitrification was not quantified here.


## 6    Temporal changes of carbonate chemistry in the OUTPACE area

Based on the available GLODAPv2 data, temporal changes in the OUTPACE area have been assessed (Fig. 5 and Table 3). The variation of oceanic parameters with time are estimated on two isopycnal layers : A layer with 25 kg m$^{-3}$ < $\sigma_\theta$ < 25,5 kg m$^{-3}$ (hereafter named $\sigma_{\theta\,25}$ ) and a layer with 27 kg m$^{-3}$ < $\sigma_\theta$ < 27.2 kg m$^{-3}$ (hereafter named $\sigma_{\theta\,27}$). These two layers correspond

to the features in $C_{ANT}$ discussed in the former section. $\sigma_{\theta\,25}$ can be considered as characteristic of the upper thermocline waters (core of the salinity maximum, Fig 2) whereas $\sigma_{\theta\,27}$ can be considered as characteristic of intermediate waters of southern origin (core of the salinity minimum). All the values associated to these two layers are spread between 145 and 301 dbar for $\sigma_{\theta\,25}$ and between 571 and 896 dbar for $\sigma_{\theta\,27}$. It must be mentioned that the study of temporal changes is based on a large sampling grid which covers the entire OUTPACE transect (see Sect. 2.5. and Fig. 1). This could add a spatial

variability that may interfere in the estimation of temporal changes.

Temporal variations of $C_T$ and $C_{ANT}$ between 1970 and 2015 are presented on Fig 5. As mentioned earlier, even if $C_{ANT}$ estimates from TrOCA could be biased, a former study by Perez et al. 2009 suggests that the TrOCA method gives similar values than other methods for estimating $C_{ANT}$ accumulation rates. A linear fit was applied to the observed temporal variations for $A_T$, $[O_2]$, $C_T$ and $C_{ANT}$ to check for significant trends on data collected between 1980 and 2015 . The results of

the performed regression analyses are presented on table 2. Trends are evaluated with and without the data of the OUTPACE cruise in order to estimate the influence of this new dataset on the observed trends. Trends are evaluated for the entire OUTPACE area and for the MA and the WGY area . Even if presented on Figure 5, data collected before 1980 from the GLODAPv2 database are disregarded in the estimation of the temporal trends. Indeed, for the OUTPACE area, data prior to 1980 originate from one single GEOSEC cruise in 1974, with only one measured point for $\sigma_{\theta\,27}$ at WGY and no points at $\sigma_{\theta\,25}$

for WGY and WMA.

At $\sigma_{\theta\,25}$, a significant decrease of $A_T$ of  -0.20 ± 0.07 µmol kg$^{-1}$.a$^{-1}$ is observed over the entire OUTPACE area. A decrease of -0.30 ± 0.09 µmol.kg$^{-1}$.a$^{-1}$ is also observed in MA area, whereas no significant trend is observed for the WGY area. However, when $A_T$ is normalized to salinity, no significant trends are observed in $A_{T\,n35}$ suggesting that the observed trend in $A_T$ can be attributed to salinity changes rather than changes in calcification. Significant negative trends are observed for $[O_2]$ over the

entire area (- 0.31 ± 0.10 µmol kg$^{-1}$ a$^{-1}$), in MA (- 0.35 ± 0.16 µmol kg$^{-1}$ a$^{-1}$) and in WGY (- 0.38 ± 0.11 µmol kg$^{-1}$ a$^{-1}$) . The decrease in $[O_2]$ wich corresponds to a positive trend in AOU suggested an increase in the remineralization of organic matter

at $\sigma_{\theta\,25}$. Significant increasing trends were observed for $C_T$ over the entire area (+ 1.32 ± 0.13 µmol kg$^{-1}$ a$^{-1}$), in MA (+ 1.38 ± 0.21 µmol kg$^{-1}$ a$^{-1}$) and in WGY (+ 1.57 ± 0.13 µmol kg$^{-1}$ a$^{-1}$) . For $C_{ANT}$, the trends were slightly slower (+ 1.12 ± 0.07 to 1.2± 0.13 ± 0.09 µmol kg$^{-1}$ a$^{-1}$) and not significantly different between MA and WGY. Taking into account the OUTPACE dataset does not change the overall significance of the observed trends and only minor changes (mostly within the error of the estimates) are observed. If we assume a $C_T$ increase of 0.5 to 1 µmol kg$^{-1}$ a$^{-1}$ (depending on the buffer factors considered) associated to the recent rise in atmospheric $CO_2$ (see for example Murata et al., 2007), the $C_T$ increase in the OUTPACE area is faster than thermodynamics would govern whereas the $C_{ANT}$ is closer to this thermodynamic value. The higher increase of $C_T$ could be related to an increase in remineralization processes as deduced from $[O_2]$ trends, with an overall consistency between the rate of $C_T$ increase and the rate of decrease in $[O_2]$. However, the important increase of $C_{ANT}$ observed between 2005 and 2015 between 10°S and 30°S on the P16 line (at the eastern side of the OUTPACE transect) by Carter et al. (2017) is not supported by significant differences in the trends of $C_{ANT}$ observed between MA and WGY in this study.

At $\sigma_{\theta\,27}$, the only significant trend observed is an increase in $C_{ANT}$ of ca. 0.40 ± 0.06 µmol.kg$^{-1}$.a$^{-1}$ in the MA area. When the OUTPACE dataset is not considered, a similar trend is observed for $C_T$ in the MA area. This trend is compatible with the observed increase of $C_{ANT}$ by Kouketsu et al. (2013) along the P21 line close to the isopycnal layer 27 kg m$^{-3}$. As this increase is not observed in WGY and if we assume that the $\sigma_{\theta\,27}$ is filed with AAIW waters, this suggest that the accumulation of $C_{ANT}$ in AAIW is faster west of 170°W line than to the east, but no clear explanation for this trend can be given.

## 7    Towards an enhanced  "Ocean Acidification" in the WTSP?

Temporal variations of pH$_T$ between 1970 and 2015 are presented on Fig. 5c and 5f with rates of pH$_T$ decrease of -0.0022 ± 0.0004 a$^{-1}$ for MA and -0.0027 ± 0.0004 a$^{-1}$ for WGY at  $\sigma_{\theta\,25}$ (Table 3) between 1980 and 2015. Based on the $C_{ANT}$ rates estimated in the previous section (1.1 to 1.2 µmol kg$^{-1}$ a$^{-1}$),  and based on a constant value of A$_T$ of 2285 µmol kg$^{-1}$ (mean value of A$_{Tn35}$ on $\sigma_{\theta\,25}$) and a constant temperature of 20°C (mean value of temperature on $\sigma_{\theta\,25}$), we can estimate a pH$_T$ decrease rate of  -0.0023 to -0.0025 a$^{-1}$. This indicates that rates of oceanic pH$_T$ decrease (ocean acidification) can mostly be explained by the increase of $C_{ANT}$. These rates of acidification are higher than the values reported by Waters et al. (2011) in the Western South Pacific along the P06 Line (south of OUTPACE area at 32°S) between two visits in 1992 and 2008. They are also higher than the surface rates of pH$_T$ decrease of –0.0016 ± 0.0001 a$^{-1}$ recorded at the HOT time-series station in the tropical North Pacific and of –0.0017 ± 0.0001 a$^{-1}$ and –0.0018 ± 0.0001a$^{-1}$ in the tropical North Atlantic at BATS and ESTOC stations respectively (Bates et al., 2014). However, differences in buffer factors between surface and subsurface can partially explain these differences. Nevertheless, our results in subsurface ($\sigma_{\theta\,25}$) based on GLODAPv2 and OUTPACE data ($C_T$ and A$_T$), are similar  to pH$_T$ trends derived from fCO2 surface observations (e.g. Lauvset et al, 2015). In the southern subtropical and equatorial Pacific regions, using SOCAT version 2, Lauvset et al. (2015) evaluate contrasting fCO2 and pH$_T$ trends, ranging between +1.1 µatm a$^{-1}$ and +3.5 µatm a$^{-1}$  for fCO2 and between -0.001 a$^{-1}$ and -0.0023 a$^{-1}$ for pH$_T$. If we revisit these estimates, using surface fCO2 observations available in the OUTPACE region (18-22°S/170-200°E) in SOCAT

version 6 (Bakker et al., 2016; www.socat.info) and assuming a constant alkalinity (2300 μmol/kg, average of surface data), we can calculate $pH_T$ and $C_T$ from fCO2 and temperature data. The resulting long-term trends for the period 1980-2016 for fCO2, $C_T$ and $pH_T$ are respectively $+1.27 \pm 0.01$ μatm a$^{-1}$, $+1.03 + \pm 0.01$ μmol kg$^{-1}$ a$^{-1}$ and $-0.0013 \pm 0.0001$ a$^{-1}$. Interestingly for the period 2000-2016 the trends are $+2.53 \pm 0.02$ μatm a$^{-1}$, $+2.02 \pm 0.02$ μmol kg$^{-1}$ a$^{-1}$ and $-0.0025 \pm 0.0003$ a$^{-1}$, suggesting an acceleration of the signals in recent years. These results based on fCO2 observations in surface waters, confirm the trends we detected for $C_T$ and $pH_T$ in subsurface layers ($\sigma_{\theta\,25}$).

On Fig. 6, estimates of the so-called "Anthropogenic $pH_T$ change" ($\Delta^{ANT}pH_T$) and "Anthropogenic $\Omega_{ara}$ change" ($\Delta^{ANT}\Omega_{ara}$) which corresponds to the difference of $pH_T$ and $\Omega_{ara}$ between the time of the OUTPACE cruise (modern time) and the pre-industrial period are presented. The $pH_T$ and $\Omega_{ara}$ correspond to the values presented on Fig. 3, whereas the pre-industrial values corresponds to $pH_T$ and $\Omega_{ara}$ estimated with $C_T$ minus $C_{ANT}$. All other parameters (temperature, salinity, alkalinity and nutrients) are assumed to remain constant over time. The main features for the distribution of $\Delta^{ANT}pH_T$ and $\Delta^{ANT}\Omega_{ara}$ logically reflect the distribution of the estimated $C_{ANT}$ in this study because $C_{ANT}$ is the only driving force in these estimations. The estimated $pH_T$ decrease reaches values slightly higher than 0.1 and the estimated $\Omega_{ara}$ decrease reaches values of 0.75 since the pre-industrial period for areas with the highest $C_{ANT}$ accumulation. When considering an error on $C_{ANT}$ of 6 μmol kg$^{-1}$, we can assume that we are able to distinguish changes of 0.0012 for $pH_T$ and 0.06 for $\Omega_{ara}$. Decreases of $pH_T$ and $\Omega_{ara}$ are thus detectable below 1000 dbar in the MA waters and above 1000 dbar in WGY waters.

A decrease of $pH_T$ of ~0.1 units since the pre-industrial period is a generally accepted value for oceanic waters affected by $C_{ANT}$ penetration (see e.g. Raven et al., 2005). Several studies have assessed the rate of ocean acidification based on successive visits to different oceanic areas. For the South Pacific Ocean, Carter et al. (2017) reports decreases of oceanic $pH_T$ since the pre-industrial period of -0.09 and -0.11 $pH_T$ units for the latitude band from 10 to 20°S and from 20 to 30°S, respectively, along the P16 line (150°W) situated on the eastern side of the OUTPACE area. These are in good agreement with our estimates in this area.

Based on an interpolation of the estimated $\Omega_{ara}$ during OUTPACE and the pre-industrial $\Omega_{ara,,}$ we calculated the depth of the horizon where $\Omega_{ara} = 1$ for the different stations of the OUTPACE transect (Table 2) in 2015 and the pre-industrial period base on the $\Delta^{ANT}\Omega_{ara}$ estimates. We observed an upward migration of aragonite saturation horizon of up to 220 m in the MA area along the OUTPACE transect (Table 2 and Fig 6c). These upward migration of the $\Omega_{ara} = 1$ horizon is higher than the migration of 30 to 100m observed between the 90$^{th}$ and the pre-industrial period in early studies (Feely et al. 2004) in the Pacific based on the WOCE dataset illustrating the continuous acidification of the WTSP.

## 8   Conclusion

Based on $A_T$, and $C_T$ data and related properties collected during the OUTPACE cruise, we estimated different parameters of the carbonate system along a longitudinal section of nearly 4000 km and up to 2000 dbar in WTSP. Even if the vertical and horizontal resolution is low compared to the WOCE lines and precludes a rigorous comparison with this high quality dataset,

we estimated that the measured carbonate chemistry parameters are in good agreement with previous data collected in this area. Based on estimation of $C_{ANT}$ from the TrOCA method, we find $C_{ANT}$ penetration in the WTSP and impacts on $pH_T$ and saturation state of calcium carbonate since the pre-industrial period that are in good agreement with previous observations in this area. As mentioned above, $C_{ANT}$ from TrOCA estimates are not reliable in surface layer. However, based on GLODAPv2

and SOCAT database, our estimation of $C_{ANT}$ in sub surface seems to be in good agreement with expected changes in surface waters. The enhanced impact of ocean acidification in the Subtropical South Pacific suggested by our study highlight the necessity of sustained research efforts in this largely under-explored part of the World Ocean. The presented dataset collected along the OUTPACE transect could complement existing section visited nearly every decade in the South Pacific ocean and in particular the P21 line which was last visited in 2009.


**Acknowledgements**

This is a contribution of the OUTPACE (Oligotrophy from Ultra-oligoTrophy PACific Experiment) project (https://outpace.mio.univ-amu.fr/) funded by the French research national agency (ANR-14-CE01-0007-01), the LEFE-CyBER program (CNRS-INSU), the GOPS program (IRD) and the CNES (BC T23, ZBC 4500048836). The OUTPACE

cruise (http://dx.doi.org/10.17600/15000900) was managed by the MIO (OSU Institut Pytheas, AMU) from Marseille (France) which has received funding from European FEDER Fund under project 1166-39417. All data and metadata are available at the following web address: http://www.obs-vlfr.fr/proof/php/outpace/outpace.php. SNAPO-CO2 service at LOCEAN is supported by CNRS-INSU and OSU Ecce-Terra. The authors thank the crew of the R/V L'Atalante for outstanding shipboard operation. Catherine Schmechtig is warmly thanked for the LEFE CYBER database management.

Aurelia Lozingot is acknowledged for the administrative work. Pierre Marrec is thanked for his insightful comments on the present work. The two anonymous referees are thanked for helping improving the manuscript.

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

**Table 1** : General description of the cast sampled for carbonate chemistry parameters during the OUTPACE cruise.

| Cast | Station | Longitude (°E) | Latitude (°N) | Time (UTC) | Max Pres. | Type* | Rosette** |
|------|---------|----------------|---------------|------------|-----------|-------|-----------|
| out_c_006 | SD 1 | 159,.9255 | -17.9418 | 2015/02/22 03:08:00 | 202 | SHAW | CLA |
| out_t_002 |  | 159.9425 | -17.9088 | 2015/02/22 07:43:00 | 2000 | INT | TMC |
| out_c_010 | SD 2 | 162.1248 | -18.6078 | 2015/02/23 00:11:00 | 199 | SHAW | CLA |
| out_c_016 |  | 162.1112 | -18.5845 | 2015/02/23 08:16:00 | 1998 | INT | CLA |
| out_c_019 | SD 3 | 165.0093 | -19.4955 | 2015/02/24 05:58:00 | 200 | SHAW | CLA |
| out_c_020 |  | 165.0082 | -19.4907 | 2015/02/24 08:14:00 | 1999 | INT | CLA |
| out_c_066 | LD A | 164.5877 | -19.2242 | 2015/03/02 14:39:00 | 200 | SHAW | CLA |
| out_c_067 |  | 164.5787 | -19.2233 | 2015/03/02 16:10:00 | 2002 | INT | CLA |
| out_c_070 | SD 4 | 168.0118 | -19.9832 | 2015/03/04 10:55:00 | 201 | SHAW | CLA |
| out_c_071 |  | 168.0157 | -19.98 | 2015/03/04 12:43:00 | 1999 | INT | CLA |
| out_c_074 | SD5 | 169.9943 | -22.0002 | 2015/03/05 08:48:00 | 201 | SHAW | CLA |
| out_c_075 |  | 169.9965 | -21.9997 | 2015/03/05 10:27:00 | 1999 | INT | CLA |
| out_c_078 | SD6 | 172.1198 | -21.3732 | 2015/03/06 07:27:00 | 200 | SHAW | CLA |
| out_c_079 |  | 172.1193 | -21.3758 | 2015/03/06 09:08:00 | 1999 | INT | CLA |
| out_c_082 | SD7 | 174.25 | -20.7697 | 2015/03/07 05:09:00 | 201 | SHAW | CLA |
| out_c_083 |  | 174.2512 | -20.7677 | 2015/03/07 06:37:00 | 2000 | INT | CLA |
| out_c_086 | SD8 | 176.3778 | -20.7027 | 2015/03/08 02:31:00 | 201 | SHAW | CLA |
| out_c_087 |  | 176.364 | -20.6945 | 2015/03/08 04:19:00 | 1997 | INT | CLA |
| out_c_091 | SD9 | 178.6087 | -20.9963 | 2015/03/09 04:57:00 | 2002 | INT | CLA |
| out_t_012 |  | 178.6062 | -20.9892 | 2015/03/09 06:46:00 | 201 | SHAW | TMC |
| out_c_094 | SD10 | -178.5105 | -20.4417 | 2015/03/10 04:10:00 | 200 | SHAW | CLA |
| out_c_095 |  | -178.5105 | -20.44 | 2015/03/10 05:48:00 | 762 | INT | CLA |
| out_c_098 | SD11 | -175.6542 | -20.0028 | 2015/03/11 00:53:00 | 207 | SHAW | CLA |
| out_c_099 |  | -175.6475 | -20.0057 | 2015/03/11 02:46:00 | 2000 | INT | CLA |
| out_c_102 | SD12 | -172.7885 | -19.5237 | 2015/03/12 00:38:00 | 200 | SHAW | CLA |
| out_c_103 |  | -172.7813 | -19.5368 | 2015/03/12 02:26:00 | 2001 | INT | CLA |
| out_c_150 | B | -170.7433 | -18.179 | 2015/03/20 12:38:00 | 204 | SHAW | CLA |
| out_c_151 |  | -170.7385 | -18.1745 | 2015/03/20 14:16:00 | 1997 | INT | CLA |
| out_c_152 | SD13 | -169.0728 | -18.2007 | 2015/03/21 10:27:00 | 501 | INT | CLA |
| out_c_163 |  | -165.9315 | -18.4282 | 2015/03/24 12:23:00 | 5027 | DEEP | CLA |
| out_c_194 | C | -165.8647 | -18.4952 | 2015/03/28 02:01:00 | 25 | REPRO | CLA |
| out_c_198 |  | -165.7915 | -18.4912 | 2015/03/28 12:42:00 | 298 | SHAW | CLA |
| out_c_199 |  | -165.7792 | -18.4842 | 2015/03/28 14:32:00 | 2001 | INT | CLA |
| out_c_209 | SD14 | -163.001 | -18.395 | 2015/03/30 05:19:00 | 300 | SHAW | CLA |
| out_c_210 |  | -162.9992 | -18.3952 | 2015/03/30 07:03:00 | 2000 | INT | CLA |
| out_c_212 | SD15 | -159.9913 | -18.265 | 2015/03/31 04:01:00 | 300 | SHAW | CLA |
| out_c_213 |  | -159.9913 | -18.2618 | 2015/03/31 05:41:00 | 2002 | INT | CLA |

*: SHAW stands for casts up to 200 dbar, INT stands for casts up to 2000m, DEEP stands for the deep cast and REPRO stands for the cast with reproducibility measurements

**: CTD rosette used for the cast. CLA is the normal CTD rosette and TMC is the trace metal clean rosette (See Sect. 2.1)

**Table 2 :** Estimated trends on $A_T$, $[O_2]$, $C_T$, $C_{ANT}$ and $pH_T$ changes in two different layers of the water column defined by isopycnal layers between 1980 and 2015 based on GLODAPv2 with (column WITH) and without (column WITHOUT) OUTPACE data added. Estimated trends are obtained from slope values of a linear regression between the studied parameters and time.

| | $25 < \sigma_\theta < 25.5$ | | $27 < \sigma_\theta < 27.2$ | |
|---|---|---|---|---|
| | WITH | WITHOUT | WITH | WITHOUT |
| Trend on $A_T$ in µmol.kg$^{-1}$.a$^{-1}$ | | | | |
| OUTPACE | -0.20 ± 0.07 (n = 167) * | -0.30 ± 0.07 (n = 142) * | -0.12 ± 0.07 (n = 180) | -0.01 ± 0.06 (n = 174) |
| MA | -0.30 ± 0.09 (n = 85) * | -0.47 ± 0.10 (n = 70) * | -0.16 ± 0.09 (n = 99) | -0.10 ± 0.09 (n = 92) |
| WGY | -0.20 ± 0.14 (n = 28) | -0.20 ± 0.19 (n = 22) | -0.20 ± 0.14 (n = 35) | -0.01 ± 0.13 (n = 31) |
| Trend on $[O_2]$ in µmol.kg$^{-1}$.a$^{-1}$ | | | | |
| OUTPACE | -0.31 ± 0.10 (n = 167)* | -0.61 ± 0.09 (n = 143)* | 0.05 ± 0.11 (n = 183) | 0.07 ± 0.10 (n = 178) |
| MA | -0.35 ± 0.16 (n = 84)* | -0.78 ± 0.17 (n = 70)* | 0.06 ± 0.11 (n = 99) | 0.04 ± 0.11 (n = 93) |
| WGY | -0.38 ± 0.11 (n = 27)* | -0.35 ± 0.14 (n = 23)* | -0.11 ± 0.30 (n = 38) | -0.22 ± 0.29 (n = 34) |
| Trend on $C_T$ in µmol.kg$^{-1}$.a$^{-1}$ | | | | |
| OUTPACE | 1.32 ± 0.13 (n = 174) * | 1.63 ± 0.13 (n = 149) * | 0.23 ± 0.13 (n = 189) | 0.27 ± 0.11 (n = 183) * |
| MA | 1.38 ± 0.21 (n = 85) * | 1.87 ± 0.21 (n = 70) * | 0.31 ± 0.16 (n = 100) | 0.44 ± 0.17 (n = 93) * |
| WGY | 1.57 ± 0.18 (n = 31) * | 1.57 ± 0.23 (n = 25) * | 0.23 ± 0.29 (n = 40) | 0.23 ± 0.29 (n = 36) |
| Trend on $C_{ANT}$ in µmol.kg$^{-1}$.a$^{-1}$ | | | | |
| OUTPACE | 1.12 ± 0.07 (n = 166) * | 1.25 ± 0.06 (n = 142) * | 0.32 ± 0.05 (n = 179) * | 0.25 ± 0.04 (n = 174) * |
| MA | 1.18 ± 0.08 (n = 84) * | 1.31 ± 0.08 (n = 70) * | 0.40 ± 0.06 (n = 98) * | 0.40 ± 0.06 (n = 92) * |
| WGY | 1.20 ± 0.09 (n = 28) * | 1.18 ± 0.10 (n = 22) * | 0.13 ± 0.09 (n = 35) | 0.11 ± 0.08 (n = 31) |
| Trend on $pH_{TINSI}$ in a$^{-1}$ | | | | |
| OUTPACE | -0.0022 ± 0.0003 (n=167)* | -0.0031 ± 0.0002 (n=142)* | -0.0001 ± 0.0003 (n=181) | -0.0002 ± 0.0002 (n=175) |
| MA | -0.0022 ± 0.0004 (n=85)* | -0.0033 ± 0.0004 (n=70)* | -0.0004 ± 0.0003 (n=100) | -0.0007 ± 0.0003 (n=93)* |
| WGY | -0.0027 ± 0.0004 (n=28)* | -0.0030 ± 0.0004 (n=22)* | -0.00008 ± 0.0006 (n=35) | -0.0007 ± 0.0006 (n=31) |

* : trend significant (p-level < 0.05)

**Table 3 :** Estimated depth of the $\Omega_{ara}$ = 1 horizon along the OUTPACE cruise (see text for details). No values are available for stations where data up to 2000 dbar were not available (SD2 and SD13). For the depth of the $\Omega_{ara}$ = 1 horizon, no values were estimated for stations with $C_{ANT}$ < - 6 µmol kg$^{-1}$.

| Station | Longitude | Latitude | Depth of the $\Omega_{ara}$ = 1 horizon (in m) | | |
|---------|-----------|----------|---------|----------|-------------|
| | | | OUTPACE | Pre-indu. | Difference* |
| SD1 | 159.9425 | -17.9088 | 1225 | NA | NA |
| SD2 | 162.1248 | -18,6078 | NA | NA | NA |
| SD3 | 165.0082 | -19.4907 | 928 | NA | NA |
| A | 164.5787 | -19.2233 | 1032 | 1185 | 153 |
| SD4 | 168.0157 | -19.98 | 1029 | 1193 | 164 |
| SD5 | 169.9965 | -21.9997 | 1126 | 1256 | 130 |
| SD6 | 172.1193 | -21.3758 | 1097 | 1233 | 136 |
| SD7 | 174.2512 | -20.7677 | 1015 | 1235 | 220 |
| SD8 | 176.364 | -20.6945 | 1010 | 1171 | 161 |
| SD9 | 178.6087 | -20.9963 | 1214 | NA | NA |
| SD11 | -175.6475 | -20.0057 | 1055 | 1172 | 117 |
| SD12 | -172.7813 | -19.5368 | 1013 | 1112 | 99 |
| B | -170.7385 | -18.1745 | 948 | 1046 | 98 |
| SD13 | -169.0728 | -18.2007 | NA | NA | NA |
| C | -165.7792 | -18.4842 | 854 | 941 | 87 |
| SD14 | -162.9992 | -18.3952 | 889 | 1006 | 117 |
| SD15 | -159.9913 | -18.2618 | 917 | 1043 | 126 |

* Difference (in m) between the depth of the $\Omega_{ara}$ = 1 horizon at the pre-industrial period and the OUTPACE cruise.

## Figures

**Fig. 1:** Map of the OUTPACE cruise transect. The outpace stations are distinguished between Melanesian Archipelago (MA) stations with darkgreen large dots and the Western GYre (WGY) stations with dark blue large dots. Stations outside of these two areas are in grey. The station with a red indication corresponds to the station where the deep cast and intercomparaison cast was made. Station from the GLODAPv2 database are indicated with small crosses: small green dots correspond to GLODAPv2 stations considered for comparaison in the MA area, small blue dots correspond to GLODAPv2 stations considered for comparaison in the WGY area and small grey dots are the other GLODAPv2 stations considered for comparison.

**Fig. 2:** $\Theta - S_A$ diagram with colors indicating the AOU. Black contour lines represent the isopycnal horizons based on potential density referenced to a pressure of 0 dBar ($\sigma_\theta$).

**Fig. 3:** Longitudinal variations of (a) $A_T$, (b) $C_T$, (c) $pH_T$ and (d) $\Omega_{ara}$ along the OUTPACE transect between surface and 2000m depth. Black contour lines represent the isopycnal horizons based on potential density referenced to a pressure of 0 dBar. Vertical profiles of (e) $A_T$, (f) $A_T$ normalized to SA =35 and (g) $C_T$ of the entire OUTPACE dataset (red dots) superimposed on the GLODAPv2 data corresponding to the OUTPACE area (grey dots).

**Fig. 4:** Longitudinal variations $C_{ANT}$ (Estimated with the TROCA method) along the OUTPACE transect between surface and 2000m depth (a). Black contour lines represents the isopycnal horizons based on potential density referenced to a pressure of 0 dBar. Vertical profiles of $C_{ANT}$ for the entire OUTPACE dataset superimposed on the values estimated from the GLODAPv2 data (b) and vertical profiles of of $C_{ANT}$ between surface and 1500m superimposed on the values estimated from the recent (after 2005) GLODAPv2 data (c). Color code for the dots is the same as for Figure 1

**Fig. 5:** Temporal evolution in the OUTPACE area of $C_T$ (a and c), $C_{ANT}$ (b and d) and $pH_{Tinsi}$ (c and e) based on GLODAPv2 and OUTPACE data along two isopycnal layers: $25 - 25.5$ kg.m$^{-3}$ (left side panels) and $27 - 27.2$ kg.m$^{-3}$ (right side panels). Color code for the dots is the same as for Figure 1.

**Fig. 6:** Longitudinal variations of (a) $pH_T$ changes and (b) $\Omega_{ara}$ changes between pre-industrial and present time along the OUTPACE transect between surface and 2000m depth (See text for details). Black contour lines represent the isopycnal horizons based on potential density referenced to a pressure of 0 dBar.

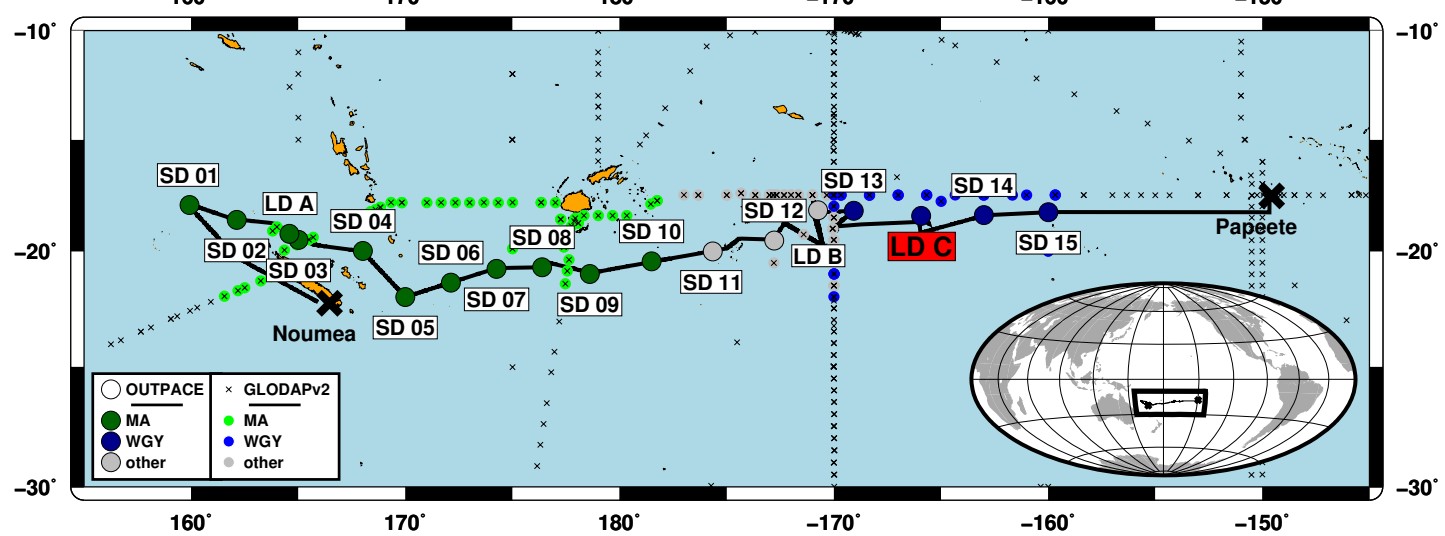

Fig. 2

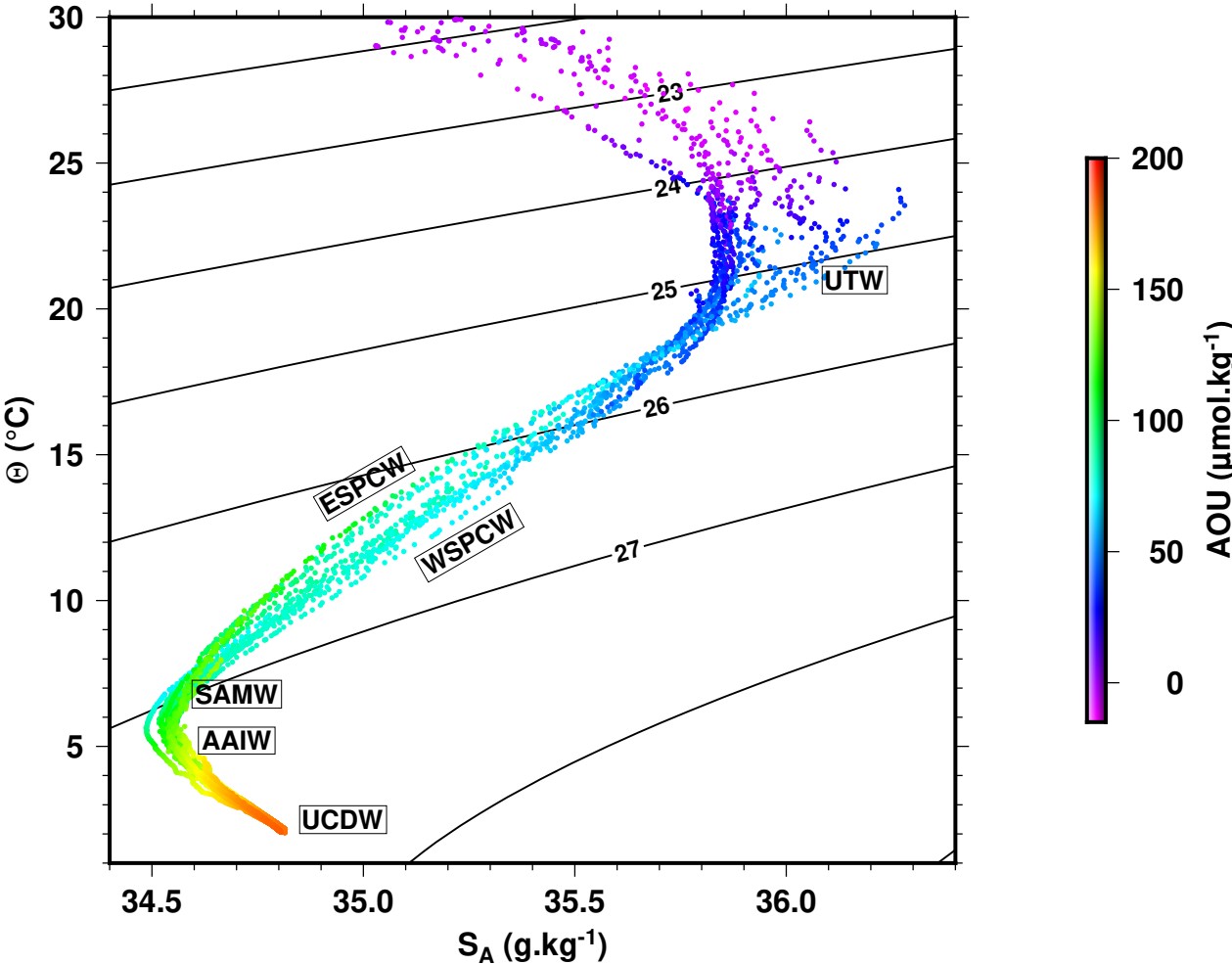

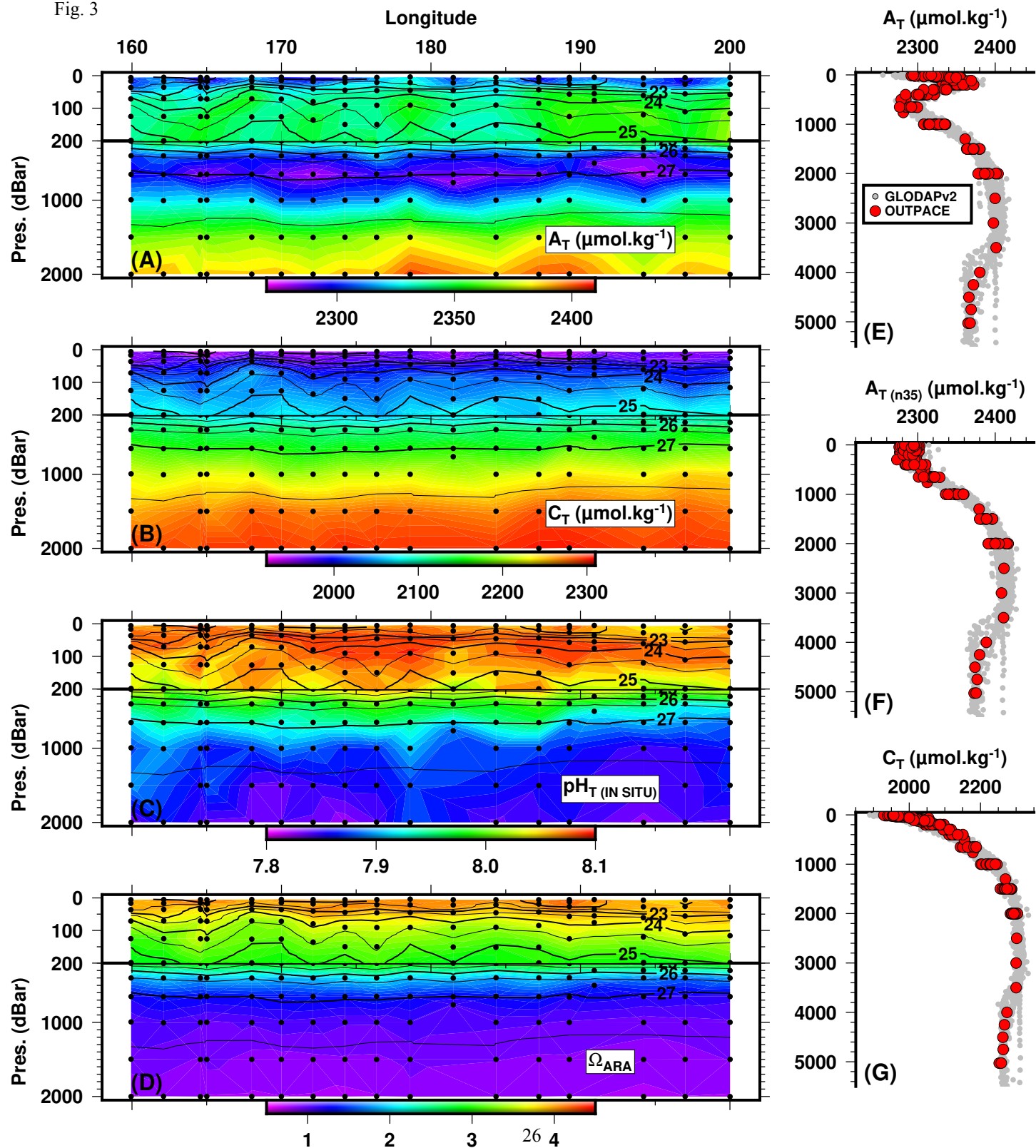

Fig. 3

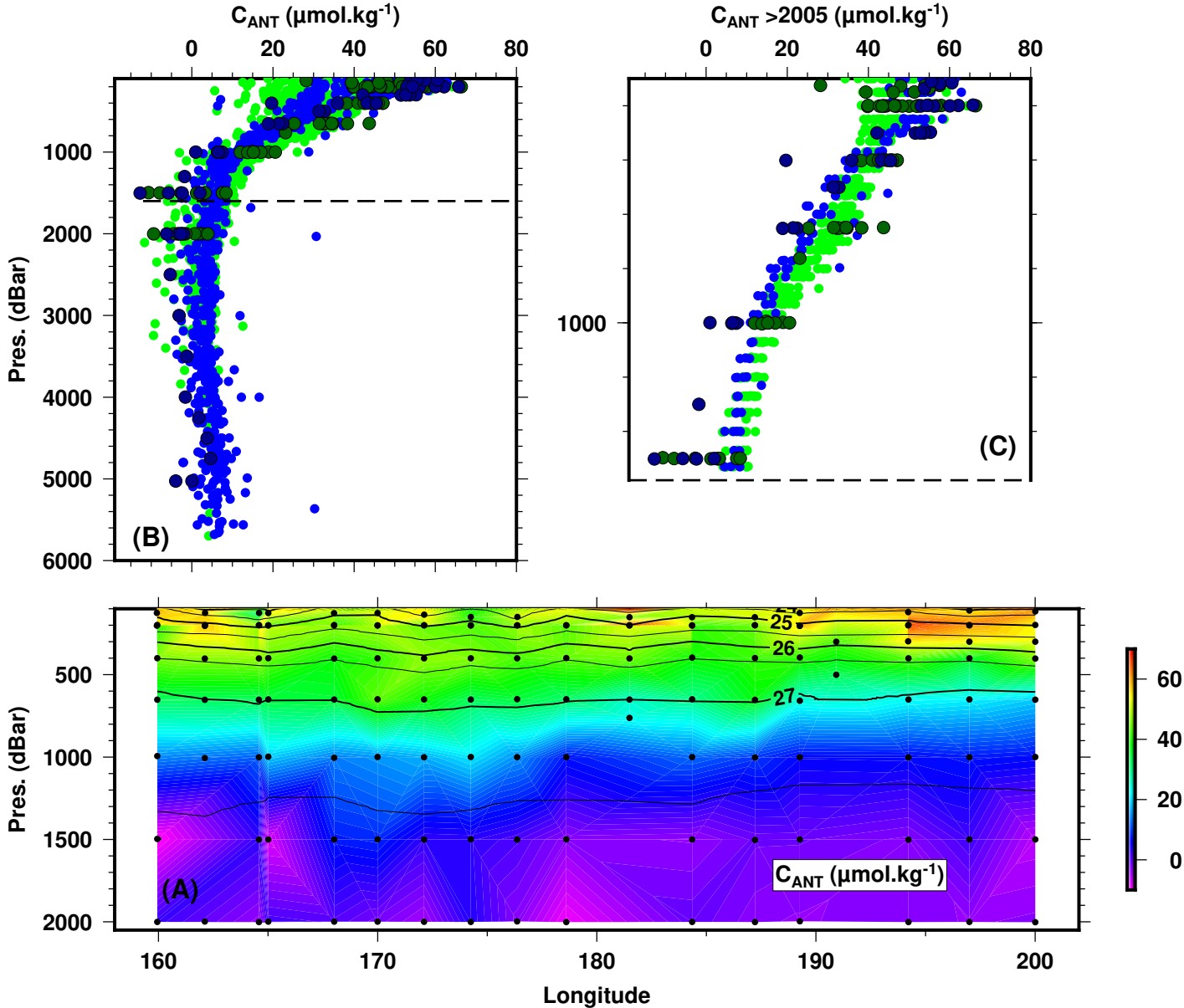

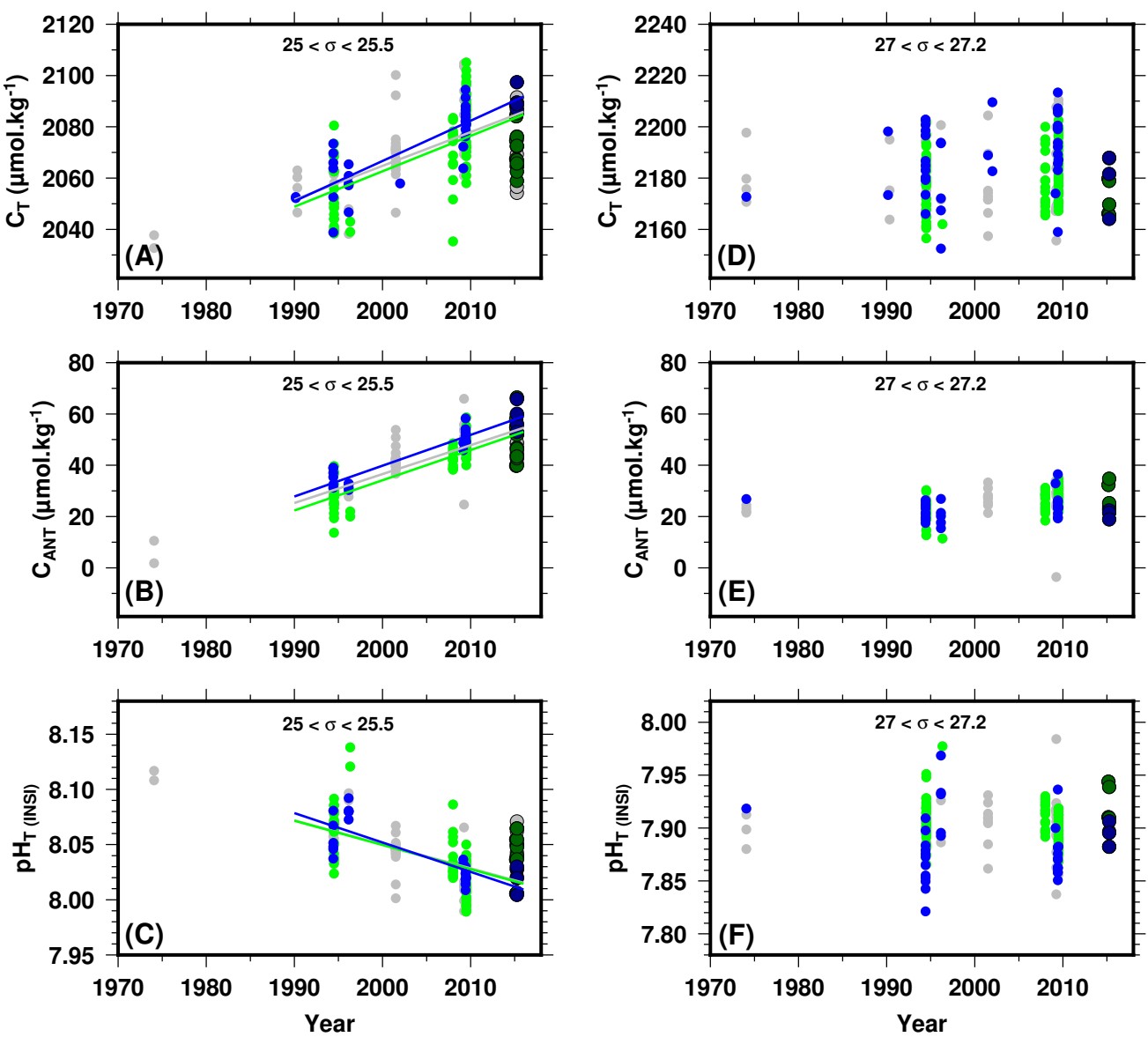

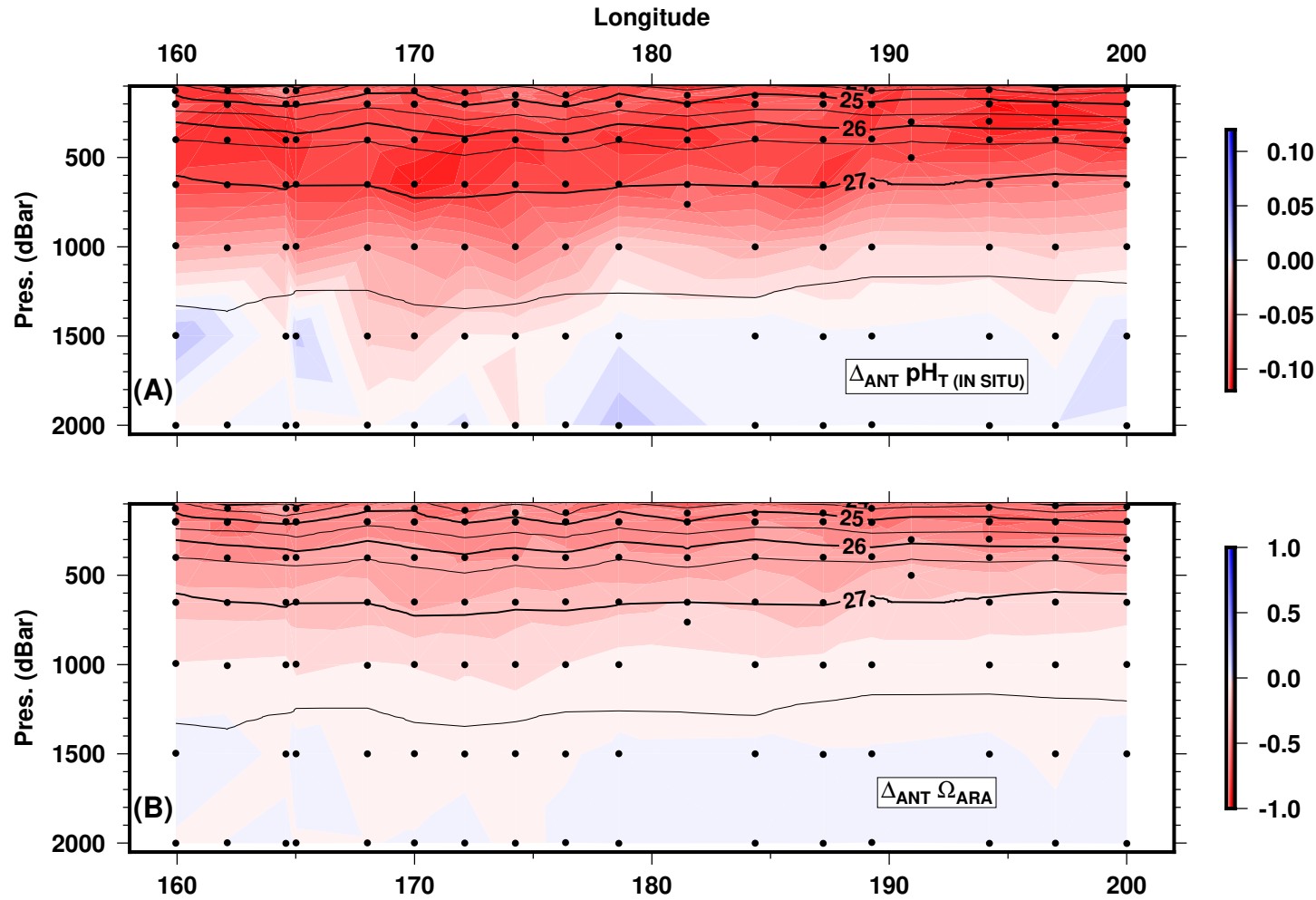