# Peer review of "Carbonate system distribution, anthropogenic carbon and acidification in the Western Tropical South Pacific (OUTPACE 2015 transect)"

_Biogeosciences, 2018_

## Referee Comment (RC1) · Anonymous Referee #1 · 3 May 2018

Review on Carbonate system distribution, anthropogenic carbon and acidification in the Western Tropical South Pacific (OUTPACE 2015 transect), by Wagener Thibaut et al. for publication in Biogeosciences Discussions.

General comments:

The main goal of the manuscript is to report a new dataset of measurements of the carbon system for the western tropical South Pacific (OUTPACE cruise). The authors describe the distribution of the different variables along the OUTPACE transect highlighting the differences between the western (Melanesian Archipelago, MA) and the eastern (western South Pacific gyre, WGY) part of the transect. The authors also present results for derived properties (pH, $\Omega_{cal}$ and $\Omega_{ara}$) of the carbon system and for anthropogenic carbon ($C_{ANT}$) that has been estimated by the TrOCA method. Making use of ancillary data, the authors present temporal changes in the properties (measured and estimated) observing: 1) a decrease in total alkalinity restricted to the MA area that disappears when using normalized alkalinity; 2) an "over accumulation" of total inorganic carbon and an increase in $C_{ANT}$ (close to the thermodynamic value) in the upper thermocline waters; 3) a decrease in pH and shoaling of the aragonite saturation depth.

The dataset reported in the study is of high quality and without any doubt complements the decadal P21 hydrographic line. The manuscript is well written and ordered and the results are well presented. Nevertheless, I have some specific comments that need to be addressed before being considered for publication.

Specific comments:

Abstract:

Page 1, line 22. Eliminate "of" after $C_{ANT}$ increases.

Page 1, line 23. "in $C_{ANT}$" instead of "of $C_{ANT}$".

Page 1, line 24. "$pH_T$" instead of "pH".

1 Introduction:

Page 2, line 42. Delete "hereafter named".

Page 2, line 54. "in recent years". Please, specify the period of study.

2.1 Cruise and sampling strategy:

Page 3, lines 83-84. "(1)" and "(2)" not needed.

2.4 Derived parameters:

Page 5, line 138. "calcite ($\Omega_{cal}$)". There is no need to mention this variable because it is not displayed in the distributions (fig. 3) and its temporal change is not estimated. See comment on section 4.

4 Carbonate chemistry along the OUTPACE transect:

Page 7, line 200. Why is $C_T$ slightly lower in bottom waters?

Page 7, line 211. "$pH_T$" instead of "pH".

Page 8, lines 217-220. Not need to add these sentences or maybe use them in section 2.4 to explain why you're not considering this variable for the temporal changes.

5 Anthropogenic carbon estimation along the OUTPACE transect:

Page 9, lines 272-279. The authors make the reader notice that denitrification could be affecting their estimates but nothing is concluded. The authors don't explain how they deal with this issue. In section 6 the authors give a reference for the low effect of denitrification over $C_{ANT}$ estimates that could be added in this section as a conclusion of why they don't consider N*.

Page 10, line 285. The year of publication of the reference is 2017.

6 Temporal changes of inorganic carbon in the OUTPACE area:

Section's title: The authors talk about other variables than just inorganic carbon. I suggest to change "inorganic carbon" to "carbonate chemistry".

Page 10, lines 303-305. Add the errors in the trends for $A_T$. What is/are the oceanic process/es behind the change/not change in alkalinity.

Page 10, lines 306-307. Add the errors in the trends for $C_T$ and $C_{ANT}$.

Page 10, line 310. Do the authors have an explanation for this "over accumulation"? What is the error in the increase of $C_T$ associated to the increase in atmospheric CO2? (line 308).

Page 11, lines 315-318. Considering the information given by the authors (page 10, lines 303-305), can the changes in AT still be due to remineralization processes? Can the authors give a possible scenario/explanation for the difference of $C_{ANT}$ between MA and WGY?

Page 11, line 319. Add the error for the change in $C_{ANT}$.

7 Towards an enhanced "Ocean Acidification" in the WTSP?:

Pages 11 and 12, lines 329, 331, 336, 337, 344, 345, 346, 347, 352, 357. "$pH_T$" instead of "pH".

Page 11, lines 341-342. Add the errors in the trends for fCO2, $C_T$ and $pH_T$.

Page 12, line 362. Add errors in the trends. They are given in the text of reference.

Page 12, lines 363-364. Add the values of the change in $\Omega_{ara}$ (with the uncertainty) that you obtained with your data. Give some explanation for the difference between your values and the ones obtained by Murata et al. (2015).

Page 12, line 368. Add the migration rate observed by Feely et al. (2004) and the period of study.

8 Conclusion:

Page 12  line 375. "pH$_T$" instead of "pH".

---

## Referee Comment (RC2) · Anonymous Referee #2 · 4 May 2018

This is a broadly well-written paper that cleanly presents the information data users would need to use the dataset being presented. In these respects, the paper is worthy of publication. However, the paper runs into trouble in the extension of its analysis to Cant. Lacking a direct earlier occupation to compare to or transient tracer information to provide ventilation age information, the dataset is ill equipped to be used for these estimates (as the authors point out). The authors therefore use the TrOCA approach for estimating Cant. TrOCA is convenient and easy to apply, but untrustworthy: https://www.biogeosciences.net/7/723/2010/

[Figure]

The authors discuss this limitation in section 5 (it should also be briefly mentioned in section 2.4), but then go on to discuss comparisons between various regions and literature estimates without further mentioning or propagating the uncertainties from the methods. This leaves the reader to believe that the uncertainties in the fits are appropriate estimates of the uncertainties in the estimates, which seems incorrect.

Fortunately, the proximity of the data set to P21 allows the authors' TrOCA estimates to be compared to earlier TrOCA estimates from 1994 and 2009. This analysis should allow much of the TrOCA methodological error to cancel when computing changes in Cant over time (see: https://doi.org/10.5194/bg-7-1789-2010, who briefly provide a bit of evidence to suggest this approach might work okay for estimating rates). However, the change in the sampling grid between the P21 cruises and this cruise could pose a separate problem for this approach (www.biogeosciences.net/10/4801/2013/), particularly when comparing regions to one another (see below).

In terms of improving the Cant discussion, I'd argue the paper should:

1. Remove the discussion of column inventories of Cant, and downplay or remove the discussion of the overall Cant distributions. Presenting these values suggests a belief in the accuracy of the values to within the stated precision of $\pm 6$ $\mu$mol kg-1 that isn't warranted given Yool et al.'s findings. Instead focus on Cant changes.

2. {Delta}nCT0 or eMLR could be used to compare the P21 section datasets to the new measurements (and to one another), useful to show that rates of change found are not byproducts of the TrOCA methods used..

3. Attempt to estimate Cant uncertainty from Yool et al., and then propagate these estimates through their calculations to estimate uncertainty in each of the values they present.

4. For dealing with the change in the sampling grid, it might be interesting to simply compare the rates found with and without the new dataset. This would allow the rates

from "P21 only" to be directly compared to Kouketsu et al. 2013, who use a different method entirely. The differences between those estimates and these could then be discussed in the context of both changes in rates and changes in sampling grids.

Alternately, the paper could likely stand as a simple presentation of the data to ESSD after removing most of the Cant discussion.

Specific comments:

13. carbonate parameters

27. recommend: 10 PgC or GtC

29. Socean/EFF=0.26 (from LeQuere), which is closer to 25% than 30%. 30% is closer to the historical average sequestration.

136. These five...

180. Consider also the "potential vorticity minimum" definition of SAMW.

207. Recommended "The estimated offsets are XXX and XXXX. These offsets are smaller than the estimated repeatability of the measurements."

- However, if this suggested text is an accurate phrasing of the idea being conveyed then it implies you believe that the repeatability is a good estimate of the potential bias... potential measurement biases on the order of the listed repeatability would completely hide decadal Cant accumulation at 95% confidence. Pehaps rephrase simply as "The estimated offsets are XXXX and XXXX, suggesting measurement biases are likely no larger.

236. depths

248. singularities should be another word... perhaps "features"

285. the high 0.8 mol C / mˆ2 estimates in Carter et al. were only for the last decade or so. The estimates in this region were smaller for the WOCE-CLIVAR period. If we

assume 1994 to 2005 with accumulations of ∼0.3 mol C /mˆ2 per year (approximated from the figure in Carter et al.) with 2005 to 2015 accumulations of 0.8 per year... it suggests a total change of ∼11 mol C /mˆ2 or so, rather than the 20 mol C /mˆ2 since 1994 found here.

298. what does it mean to be adjusted to a linear model? Possible recommended rephrase: "a line was fit to the data. . ."

300. what did they distinguish?

305. trends

312. I do not understand this sentence. How could Cant accumulation be related to denitrification? Denitrification does not change Cant.

319. The change in the sampling grids means you can't trust these linear fits in the MA region. Your measurements in MA are south of the P21 section where, being closer to the ventilation regions for AAIW and SAMW, you would expect higher Cant. The fact that your measurements are higher relative to P21 here than elsewhere is potentially attributable to that alone.

325 is -> are

Section 7. Here the authors compare their subsurface pHT changes to some surface pHT changes in literature. These are not valid comparisons because subsurface Cant is frequently lower and because the impact of Cant on pH is increased in the surbsurface where Revelle factors are higher.

328. why 20C?

376. observations

Table 3. Commas are used for decimal points at places in this manuscript while periods are used in other places.

Figures. The section plots are tessellated (faint lines going everywhere on them), which is a problem that seems to happen for Matlab 2014b+ when exporting to vector graphics. Consider exporting to high resolution raster files instead. Please ignore if this is just a function of the review-proof.

Colormaps: With the exception of Figure 6 (which would be impossible for people who are red-green colorblind to read), there are no changes to the colormaps that need to be made for this paper to be publishable. However, the authors should give this resource a read:

https://matplotlib.org/cmocean/

At the end of the webpage there are links to papers making the case that rainbow colormaps are not ideal for communicating science. The rest of the page is dedicated to providing alternatives.

---

## Author Comment (AC1) · 20 Jul 2018

**Response to the Interactive comment by « Anonymous Referee #1 » on "Carbonate system distribution, anthropogenic carbon and acidification in the Western Tropical South Pacific (OUTPACE 2015 transect)".**

**Please note that the referee comments are typesetted with** normal characters **and our responses to referee's comments are in bold characters. Text from the manuscript is indicated with** *italic characters* **and changes are highlighted** *in red***. For minor changes mentioned by the referee, we sometimes just mention that we agree with the referee and we will of course make the corrections in the revised manuscript.**

General comments:
The main goal of the manuscript is to report a new dataset of measurements of the carbon system for the western tropical South Pacific (OUTPACE cruise). The authors describe the distribution of the different variables along the OUTPACE transect highlighting the differences between the western (Melanesian Archipelago, MA) and the eastern (western South Pacific gyre, WGY) part of the transect.
The authors also present results for derived properties (pH, $\Omega$ cal and $\Omega$ ara ) of the carbon system and for anthropogenic carbon ($C_{ANT}$ ) that has been estimated by the TrOCA method. Making use of ancillary data, the authors present temporal changes in the properties (measured and estimated) observing: 1) a decrease in total alkalinity restricted to the MA area that disappears when using normalized alkalinity; 2) an "over accumulation" of total inorganic carbon and an increase in $C_{ANT}$ (close to the thermodynamic value) in the upper thermocline waters; 3) a decrease in pH and shoaling of the aragonite saturation depth.
The dataset reported in the study is of high quality and without any doubt complements the decadal P21 hydrographic line. The manuscript is well written and ordered and the results are well presented.
Nevertheless, I have some specific comments that need to be addressed before being considered for publication.
**First, we would like to thank Referee #1 for his/her careful evaluation of our manuscript. We believe that his/her comments will help to improve the manuscript. Please, find hereafter our responses to the concerns raised by Referee #1**

Specific comments:

Abstract:
Page 1, line 22. Eliminate "of" after C ANT increases.
**We agree with this correction.**
Page 1, line 23. "in C ANT " instead of "of C ANT ".
**We agree with this correction.**
Page 1, line 24. "pH T " instead of "pH".
**We agree with this correction.**

1 Introduction:
Page 2, line 42. Delete "hereafter named".
**We agree with this correction.**

2.4 Derived parameters:
Page 5, line 138. "calcite ($\Omega$ cal )". There is no need to mention this variable because it is not displayed in the distributions (fig. 3) and its temporal change is not estimated. See comment on section 4.
**The reference to ($\Omega$ cal ) estimates was deleted but a sentence has been added to justify why this parameter was not considered.**

*Seawater pH on the total scale ($pH_T$) and the $CaCO_3$ saturation state with respect to aragonite ($\Omega_{ara}$) were derived from $A_T$ and $C_T$ with the "Seacarb" R package (Gattuso and Lavigne, 2009). CaCO_3 saturation state with respect to calcite was not considered because seawater up to 2000 dbar was supersaturated with respect to calcite ($\Omega_{cal}>1$).*

4 Carbonate chemistry along the OUTPACE transect:
Page 7, line 200. Why is C T slightly lower in bottom waters?
**A possible explanation is that in the South Pacific, the deep waters are among the oldest waters in the world ocean with high carbon content whereas the northward moving bottom waters have not had the time to accumulate as much carbon (see for example Murata et al. 2007). The sentence has been modified as follow:**
*The $C_T$ gradient in the upper water column has been described in Moutin et al. (2008). Below 2000 dbar, $C_T$ is relatively invariant with slightly lower values in the bottom waters (below 4000 dbar) due to the presence of very old deep waters originating from the north Pacific relative to the northward moving bottom waters that have not accumulated as much carbon (Murata et al. 2007 ).*

Page 7, line 211. "pH T " instead of "pH".
**We agree with this correction.**
Page 8, lines 217-220. Not need to add these sentences or maybe use them in section 2.4 to explain why you're not considering this variable for the temporal changes.
**These sentences have been deleted and a sentence has been added earlier in section 2.4 to justify that $\Omega_{cal}$ will not be considered.**

5 Anthropogenic carbon estimation along the OUTPACE transect:
Page 9, lines 272-279. The authors make the reader notice that denitrification could be affecting their estimates but nothing is concluded. The authors don't explain how they deal with this issue. In section 6 the authors give a reference for the low effect of denitrification over C ANT estimates that could be added in this section as a conclusion of why they don't consider N*.
**We agree with this comment of the referee. Based on the suggestions of the referee, we have rephrased this section in order to be clearer.**
*Finally, it should also be mentioned that, due to the presence of one of the main OMZ area, denitrification occurs in the eastern South Pacific and can be traced by the N* parameter (Gruber and Sarminento, 2007). Denitrification, by transforming organic carbon to inorganic carbon without consumption of oxygen, could induce an overestimation of $C_{ANT}$ by the TrOCA method (and other back calculation methods) due to a biological release of $C_T$ that is not taken into account in the formulation of the quasi conservative TrOCA tracer. Horizontal advection by the south equatorial current of the strong negative N* signal originating from the Eastern Pacific towards the western Pacific was previously described (Yoshikawa et al., 2015). Fumenia et al. (2018) have estimated N* along the OUTPACE transect and show slightly negative N* values in the upper thermocline waters at the eastern side of the OUTPACE transect where the highest CANT values are estimated. However, Murata et al. (2007) showed that, based on a direct relation between $C_T$ and N*, the influence of denitrification should be negligible on $C_{ANT}$ estimations in this area. Therefore, the N* correction has not been introduced in the $C_{ANT}$ estimates and the effect of denitrication was not quantified here.*

Page 10, line 285. The year of publication of the reference is 2017.
**We agree with this correction. However, following a suggestion of referee #2, this section will be deleted in the revised manuscript.**

6 Temporal changes of inorganic carbon in the OUTPACE area:
Section's title: The authors talk about other variables than just inorganic carbon. I suggest to change "inorganic carbon" to "carbonate chemistry".

**We agree with this correction.**

Page 10, lines 303-305. Add the errors in the trends for A T . What is/are the oceanic process/es behind the change/not change in alkalinity.

**Errors have been added for $A_T$ trends. Concerning the main drivers of $A_T$ changes in the ocean (Wolf-Gladrow et al. 2007): The major change in $A_T$ can be attributed to changes in major conservative cations and anions (i.e. salinity). The other important changes in $A_T$ are due to the biological precipitation of calcium carbonate and/or the dissolution of biogenic calcium carbonate. Finally, minor changes in $A_T$ can be attributed to biological assimilation and remineralization of nitrate. $A_T$ in the ocean is not affected by changes in the $CO_2$ content of the ocean. In our study, when $A_T$ is normalized to salinity, no significant trends in $A_T$ n35 are observed, suggesting that the observed trends in $A_T$ can be attributed to salinity changes. The manuscript has been changed as follow:**

*However, when $A_T$ is normalized to salinity, no significant trends are observed in $A_{T\,n35}$ suggesting that the observed trends in $A_T$ can be attributed to changes in salinity rather than in calcification.*

Page 10, lines 306-307. Add the errors in the trends for C T and C ANT .
**We agree with this correction.**

Page 10, line 310. Do the authors have an explanation for this "over accumulation"? What is the error in the increase of C T associated to the increase in atmospheric CO2? (line 308).
**We agree with the referee that the discussion on this "over accumulation" was not precise enough. In the revised manuscript we will rephrase this section as mentioned in the next comment.**

Page 11, lines 315-318. Considering the information given by the authors (page 10, lines 303-305), can the changes in AT still be due to remineralization processes? Can the authors give a possible scenario/explanation for the difference of C ANT between MA and WGY?
**As mentioned in the above comment, this section has been rewritten as follow:**

*At $\sigma_{\theta\,25}$, a significant decrease of $A_T$ of  -0.20 ± 0.07 µmol kg$^{-1}$.a$^{-1}$ is observed over the entire OUTPACE area. A decrease of -0.30 ± 0.09 µmol.kg$^{-1}$.a$^{-1}$ is also observed in the MA area, whereas no significant trend is observed for the WGY area. However, when $A_T$ is normalized to salinity, no significant trends are observed in $A_{T\,n35}$ suggesting that the observed trend in $A_T$ can be attributed to salinity changes rather than changes in calcification. Significant negative trends are observed for $[O_2]$ over the entire area (- 0.31 ± 0.10 µmol kg$^{-1}$ a$^{-1}$) with - 0.35 ± 0.16 µmol kg$^{-1}$ a$^{-1}$ in the MA and - 0.38 ± 0.11 µmol kg$^{-1}$ a$^{-1}$ in the WGY. The decrease in $[O_2]$ which corresponds to a positive trend in AOU suggested an increase in the remineralization of organic matter at $\sigma_{\theta\,25}$ . Significant increasing trends were observed for $C_T$ over the entire area (+ 1.32 ± 0.13 µmol kg$^{-1}$ a$^{-1}$), in the MA (+ 1.38 ± 0.21 µmol kg$^{-1}$ a$^{-1}$) and in the WGY (+ 1.57 ± 0.13 µmol kg$^{-1}$ a$^{-1}$). For $C_{ANT}$, the trends were slightly slower (+ 1.12 ± 0.07 to 1.2± 0.13 ± 0.09 µmol kg$^{-1}$ a$^{-1}$) and not significantly different between the MA and the WGY. Taking into account the OUTPACE dataset does not change the overall significance of the observed trends and only minor changes (mostly within the error of the estimates) are observed. If we assume a $C_T$ increase of 0.5 to 1 µmol kg$^{-1}$ a$^{-1}$ (depending on the buffer factors considered) associated to the recent rise in atmospheric $CO_2$ (see for example Murata et al., 2007), the $C_T$ increase in the OUTPACE area is faster than thermodynamics would govern whereas the $C_{ANT}$ is closer to this thermodynamic value. The higher increase of $C_T$ could be related to an increase in remineralization as deduced from  $[O_2]$ trends, with an overall consistency between the rate of $C_T$ increase and the rate of $[O_2]$ decrease. Howerver, the important increase of $C_{ANT}$ observed between 2005 and 2015 between 10°S and 30°S on the P16 line (at the eastern side of the OUTPACE transect) by Carter et al. (2017) is not supported by significant differences in the trends of $C_{ANT}$ observed between MA and WGY in this study.*

Page 11, line 319. Add the error for the change in C ANT.

**We agree with this correction.**

7 Towards an enhanced "Ocean Acidification" in the WTSP?:
Pages 11 and 12, lines 329, 331, 336, 337, 344, 345, 346, 347, 352, 357. "pH T " instead of "pH".
**We agree with this correction.**

Page 11, lines 341-342. Add the errors in the trends for fCO2, C T and pH T .
**We agree with this correction.**

Page 12, line 362. Add errors in the trends. They are given in the text of reference.
**We agree with this correction.**

Page 12, lines 363-364. Add the values of the change in $\Omega$ ara (with the uncertainty) that you obtained with your data. Give some explanation for the difference between your values and the ones obtained by Murata et al. (2015).
**This section was probably unclear. The aim of this section was to discuss our estimates of "anthropogenic $\Omega_{ara}$ change" since the preindustrial period. Indeed, we do not discuss decadal $\Omega_{ara}$ changes which were not estimated here. The reason why we compared with the Murata et al. study was to point out the interesting longitudinal differences in the $\Omega_{ara}$ decrease observed in the recent years (1994 to 2009) in the OUTPACE area which are attributed, at least partially, to changes in sea surface temperature, that we do not observe on our long term estimates. However, we believe this section was confusing for the reader and we will removed this comparison with Murata et al. (2015).**

Page 12, line 368. Add the migration rate observed by Feely et al. (2004) and the period of study.
**In the study by Feely et al (2004), upward migration of $\Omega_{ara}$ horizons between the preindustrial period and present (late 90s) are evaluated by a method comparable to ours and values between 30 and 100 m are given for the Pacific Ocean. These values will be added to the manuscript.**

8 Conclusion:
Page 12 line 375. "pH T " instead of "pH".
**We agree with this correction**

---

## Author Comment (AC2) · 20 Jul 2018

**Response to the Interactive comment by « Anonymous Referee #2 » on "Carbonate system distribution, anthropogenic carbon and acidification in the Western Tropical South Pacific (OUTPACE 2015 transect)".**

**Please note that the referee comments are typesetted with** normal characters **and our responses to referee's comments are in bold characters. Text from the manuscript is indicated with** *italic characters* **and changes are highlighted** *in red***. For minor changes mentioned by the referee, we sometimes just mention that we agree with the referee and we will of course make the corrections in the revised manuscript.**

This is a broadly well-written paper that cleanly presents the information data users would need to use the dataset being presented. In these respects, the paper is worthy of publication. However, the paper runs into trouble in the extension of its analysis to Cant. Lacking a direct earlier occupation to compare to or transient tracer information to provide ventilation age information, the dataset is ill equipped to be used for these estimates (as the authors point out). The authors therefore use the TrOCA approach for estimating Cant. TrOCA is convenient and easy to apply, but untrustworthy: The authors discuss this limitation in section 5 (it should also be briefly mentioned in section 2.4), but then go on to discuss comparisons between various regions and literature estimates without further mentioning or propagating the uncertainties from the methods. This leaves the reader to believe that the uncertainties in the fits are appropriate estimates of the uncertainties in the estimates, which seems incorrect. Fortunately, the proximity of the data set to P21 allows the authors' TrOCA estimates to be compared to earlier TrOCA estimates from 1994 and 2009. This analysis should allow much of the TrOCA methodological error to cancel when computing changes in Cant over time (see: https://doi.org/10.5194/bg-7-1789-2010, who briefly provide a bit of evidence to suggest this approach might work okay for estimating rates). However, the change in the sampling grid between the P21 cruises and this cruise could pose a separate problem for this approach (www.biogeosciences.net/10/4801/2013/), particularly when comparing regions to one another (see below).

In terms of improving the Cant discussion, I'd argue the paper should:
1. Remove the discussion of column inventories of Cant, and downplay or remove the discussion of the overall Cant distributions. Presenting these values suggests a belief in the accuracy of the values to within the stated precision of ±6 μmol kg-1 that isn't warranted given Yool et al.'s findings. Instead focus on Cant changes.
2. {Delta}nCT0 or eMLR could be used to compare the P21 section datasets to the new measurements (and to one another), useful to show that rates of change found are not byproducts of the TrOCA methods used.
3. Attempt to estimate Cant uncertainty from Yool et al., and then propagate these estimates through their calculations to estimate uncertainty in each of the values they present.
4. For dealing with the change in the sampling grid, it might be interesting to simply compare the rates found with and without the new dataset. This would allow the rates from "P21 only" to be directly compared to Kouketsu et al. 2013, who use a different method entirely. The differences between those estimates and these could then be discussed in the context of both changes in rates and changes in sampling grids.
Alternately, the paper could likely stand as a simple presentation of the data to ESSD after removing most of the Cant discussion.

**First of all, we would like to thank referee #2 for his/her careful evaluation of our manuscript. We believe that his/her comments will help to improve the manuscript. We would also like to acknowledge his/her frank but courteous criticism on our estimates of $C_{ANT}$ by the TrOCA method. We understand that the reviewer does not agree with our estimates of $C_{ANT}$ because (s)he considers the TrOCA method untrustworthy. We are aware of the scientific debate that**

exists on the different methods to estimate $C_{ANT}$ in the water column and in particular the TrOCA method.

We believe that this manuscript should be part of the "Interactions between planktonic organisms and biogeochemical cycles across trophic and N2 fixation gradients in the western tropical South Pacific Ocean: a multidisciplinary approach (OUTPACE experiment)" special issue in Biogeosciences. As mentioned in the manuscript, even if the dataset presented has been partially used in Moutin et al. 2018 (this issue), we consider that this manuscript gives a complete information on the carbonate data acquired during the OUTPACE cruise. Comparing the "OUTPACE" data to the GLODAPv2 dataset confirms recent trends observed in the South Pacific in terms of increase of the carbon content and ocean acidification. However, as mentioned by the referee, our dataset "is ill equipped" to be used for $C_{ANT}$ estimates because the OUTPACE transect does not correspond to a earlier occupation in the South Pacific and no transient tracer information to provide ventilation age information is available. Moreover, the horizontal and vertical resolution of our dataset is relatively low and for most of the stations no data have been collected below 2000 dBar. This makes our dataset less extensive compared to the earlier "WOCE lines" cruises that took place in this area (e.g. Carter et al. 2017, Kouketsu et al. 2013).

Despite the limitations of the TrOCA method (which are explicitly exposed in the manuscript, see response below), we decided to apply it because of its simplicity in the case of our dataset. Even if the TrOCA method could produce some wrong estimates of $C_{ANT}$, we believe that it can be used as a tool to investigate changes in $C_T$ content and that it can be valuable for estimating $C_{ANT}$ accumulation rates.

In order to respond to the general concerns of referee 2, we propose the following changes in the manuscript:

1. We will remove our estimates of $C_{ANT}$ column inventories. We will delete lines 251 to 254, lines 280 to 288, line 376 and lines 380 to 382 of the manuscript. We will also remove the first panel of Table 2 (which will become Table 3) and we will remove the $C_{ANT}$ inventories on figure 6 (last panel). The new Figure 6 and Table 3 can be seen at the end of this document. We propose to modify lines 18 to 20 of the abstract as follow:

*Along the OUTPACE transect, a deeper penetration of $C_{ANT}$ in the intermediate waters was observed in the MA, whereas highest $C_{ANT}$ concentrations were detected in the sub-surface waters of the WGY.*

2. We will expose more clearly the motivations and limitations of our dataset and describe more clearly what we want to state with our manuscript. We propose the following changes to the introduction:

*The aim of this paper is to report a new dataset of oceanic inorganic carbon (based on measurements of $C_T$ and total alkalinity ($A_T$) ) acquired in the WTSP during the OUTPACE (Oligotrophic to UltTra oligotrophic PACific Experiment) cruise performed in 2015 (Moutin et al., 2017). The main focus of the OUTPACE cruise was to study the complex interactions between planktonic organisms and the cycle of biogenic elements on different scales, motivated by the fact that the WTSP has been identified as a hot spot of $N_2$ fixation (Bonnet et al., 2017). The data presented here have been partially used in another paper of the special issue (Moutin et al., 2018) in order to study the biological carbon pump in the upper (surface to 200m) water column. In this paper we will explore the carbonate data between the surface and 2000 m depth. The OUTPACE transect (Figure 1) is close to existing WOCE and GO-SHIP lines in the South Pacific : it is parallel to the zonal P21 line (18° S visited in 1994 and 2009) and the P06 line (32° S visited in 1992, 2003 and 2010), it is crossed by the meridional P14 line (180° E visited in 1994 and 2007) and P15 line (170° W visited in 2001, 2009 and 2016) and it is situated at the eastern side of the P16 line (150° W visited in 1992, 2005 and 2014). However, the OUTPACE transect does not correspond to any earlier occupation of the "WOCE lines" in the South Pacific and no tracers of water mass age were measured during the cruise, which limits the possibilities of a robust analysis of $C_{ANT}$ accumulation in the area. Moreover, the horizontal and vertical resolution of the OUTPACE*

*dataset is low. In consequence, the OUTPACE dataset cannot be used to look at decadal changes in $C_{ANT}$ content in the South Pacific (e.g. Carter et al. 2017, Kouketsu et al. 2013). Here, $C_{ANT}$ estimates based on the TrOCA (Tracer combining Oxygen, inorganic Carbon and total Alkalinity) method will be used as a tool to investigate changes in $C_T$. Moreover,* by comparing our data with the high quality data (internally consistent through a secondary quality control (Olsen et al., 2016)) available in the Global Ocean Data analysis Project version 2 (GLODAPv2 database), will allow to evaluate $C_T$, $A_T$, $C_{ANT}$ (for TrOCA) and $pH_T$ (pH on total scale) trends in sub surface waters and at depth.

**3. We will mention in the Sect. 2.4 the limitation of the $C_{ANT}$ estimates by the TrOCA method and we will refer to the section 5 for a longer discussion on these limitations. We propose the following changes to the Sect. 2.4 :**

*This formulation is based on an adjustment of the TrOCA coefficients using $\Delta^{14}C$ and CFC-11 from the GLODAP-V1 database (Key et al., 2014). Touratier et al. (2007) estimated the overall uncertainty of the $C_{ANT}$ with TrOCA method to ca. 6 µmol kg⁻¹ based on the random propagation of the uncertainties on the variables ($C_T$, $A_T$, $[O_2]$ and θ) and coefficients used in Eq. 1. The limitations and validity of the TrOCA method will be discussed in detail in Sect. 5.*

**4. We will expose more clearly the limitations of the TrOCA method in section 5 and we will give estimates of the error associated to the TrOCA method based on the study by Yool et al. 2009. We propose following changes to the Sect. 5 :**

*As no tracers of water mass age were measured during the OUTPACE cruise, the main motivation for using the TrOCA method was to make $C_{ANT}$ estimations based on a simple calculation from parameters acquired within the cruise as done in other cruises conducted in south tropical Pacific waters (e.g. Azouzi et al., 2009; Ganachaud et al., 2017). Even if $C_{ANT}$ estimates from TrOCA could be biased, the application of a simple back-calculation method that accounts for biologically induced relative changes in $C_T$ is used here to identify some spatial features in the distribution of the carbonate system along the OUTPACE transect. Based on Yool et al. (2010), the error on the TrOCA $C_{ANT}$ estimates will be considered here as the normalized standard deviation of 1.67 for the TrOCA variant optimized with world ocean data (See Table 2 in Yool et al. 2010).*

**6. In section 6, based on the suggestion of the Referee, we will estimate the trends with and without the OUTPACE data in order to illustrate that the OUTPACE data confirms the trends observed with GLODAP_v2. However, we believe that the differences between those two estimates cannot illustrate the influence of changes in sampling grids because the estimation of the trend is not only based on P21 data but on all available data in the OUTPACE area. We will change Table 3 (that will become Table 2) with the estimates of the trends with and without the OUTPACE data. In addition to the new Table 2 (that can be seen at the end of this document), we propose the following changes to the Sect. 6 :**

*Based on the available GLODAPv2 data, temporal changes in the OUTPACE area have been assessed (Fig. 5 and Table 3). The variation of oceanic parameters with time are estimated on two isopycnal layers : A layer with 25 kg m⁻³ < σ_θ < 25,5 kg m⁻³ (hereafter named σ_θ 25) and a layer with 27 kg m⁻³ < σ_θ < 27.2 kg m⁻³ (hereafter named σ_θ 27). These two layers correspond to the features in $C_{ANT}$ discussed in the former section. σ_θ 25 can be considered as characteristic of the upper thermocline waters (core of the salinity maximum, Fig 2) whereas σ_θ 27 can be considered as characteristic of intermediate waters of southern origin (core of the salinity minimum). All the values associated to these two layers are spread between 145 and 301 dbar for σ_θ 25 and between 571 and 896 dbar for σ_θ 27. It must be mentioned that the study of temporal changes is based on a large sampling grid which covers the entire OUTPACE transect (see Sect. 2.5. and Fig. 1). This could add a spatial variability that may interfere in the estimation of temporal changes.*

*Temporal variations of $C_T$ and $C_{ANT}$ between 1970 and 2015 are presented on Fig 5. As mentioned earlier, even if $C_{ANT}$ estimates from TrOCA could be biased, a former study by Perez et al. 2009 suggests that the TrOCA method gives similar values than other methods for estimating $C_{ANT}$ accumulation rates. A linear fit was applied to the observed temporal variations* for $A_T$, $[O_2]$, $C_T$ and $C_{ANT}$ to check for significant trends on data collected between 1980 and 2015 (OUTPACE*

*cruise). The results of the performed regression analyses are presented on table 2. Trends are evaluated with and without the OUTPACE cruise data in order to estimate the influence of this new dataset on the observed trends. Trends are evaluated for the entire OUTPACE area and for the MA and the WGY areas. Even if presented on Figure 5, data collected before 1980 from the GLODAPv2 database are disregarded in the estimation of the temporal trends. Indeed, for the OUTPACE area, data prior to 1980 originates from one single GEOSEC cruise in 1974, with only one measured point at $\sigma_{\theta\,27}$ for WGY and no points at $\sigma_{\theta\,25}$ for WGY and MA.*

**7. We will change Figure 1. We will only present the GLODAP V2 data that have been used in this study and not all data that exists in the area covered in the map. We believe that this might have been confusing in the manuscript. The new Figure 1 can be seen at the end of this document.**

**Finally, we think that a complementary analysis of $C_{ANT}$ based on other methods as suggested by the referee would be out of the scope of this study and would completely modify the objectives that we assigned to this manuscript. Moreover such a complementary analysis could be difficult to realize due to the reasons mentioned earlier.**

Specific comments:
13. carbonate parameters
**We agree with this correction.**
27. recommend: 10 PgC or GtC
**We changed $10^{13}$ kg to 10 PgC.**

29. Socean/EFF=0.26 (from LeQuere), which is closer to 25% than 30%. 30% is closer
to the historical average sequestration.
**We agree with this comment and we will change 30% to 25% to more correctly represent the estimate in Le Quéré et al. (2018).**

136. These five. . .
**We agree with this correction.**

180. Consider also the "potential vorticity minimum" definition of SAMW.
**We will change this sentence as follow :**
*Hartin et al. (2011) defines SAMW with $\sigma_\theta$ values between 26.80 and 27.06 kg m$^{-3}$ corresponding to a minimum in potential vorticity, and AAIW with $\sigma_\theta$ values between 27.06 and 27.40 kg m$^{-3}$.*

207. Recommended "The estimated offsets are XXX and XXXX. These offsets are smaller than the estimated repeatability of the measurements."
- However, if this suggested text is an accurate phrasing of the idea being conveyed then it implies you believe that the repeatability is a good estimate of the potential bias… potential measurement biases on the order of the listed repeatability would completely hide decadal Cant accumulation at 95% confidence. Pehaps rephrase simply as "The estimated offsets are XXXX and XXXX, suggesting measurement biases are likely no larger.
**We understand this concern of the reviewer. We rephrased the sentence following his/her last suggestion :** *"The estimated offsets are -2.0 ± 4.2 µmol kg$^{-1}$ for $A_T$ and -2.0 ± 4.4 µmol kg$^{-1}$ for $C_T$ suggesting measurement biases are likely no larger"***.**

236. depths
**We agree with this correction.**

248. singularities should be another word. . . perhaps "features"

**We have changed "singularities" to "features".**

285. the high 0.8 mol C / m^2 estimates in Carter et al. were only for the last decade or so. The estimates in this region were smaller for the WOCE-CLIVAR period. If we assume 1994 to 2005 with accumulations of ∼0.3 mol C /m^2 per year (approximated from the figure in Carter et al.) with 2005 to 2015 accumulations of 0.8 per year... it suggests a total change of ∼11 mol C /m^2 or so, rather than the 20 mol C /m^2 since 1994 found here.
**As mentioned earlier this section on C$_{ANT}$ inventories has been removed from the manuscript. However we agree that combining the estimates "WOCE-CLIVAR" from Sabine et al. 2008 to the CLIVAR-GOSHIP estimates from Carter et al. 2017 was inconsistent.**

298. what does it mean to be adjusted to a linear model? Possible recommended rephrase: "a line was fit to the data. . ."
**We agree that the sentence "to be adjusted to a linear model" was unappropriated and was replaced by "a linear fit was applied to..." . See point 6 in the general response.**

300. what did they distinguish?
**This sentence was confusing and has been rephrased. See point 6 in the general response.**

305. trends
**We agree with this correction**

312. I do not understand this sentence. How could Cant accumulation be related to denitrification? Denitrification does not change Cant.
**We agree that denitrification does not change C$_{ANT}$ accumulation, but we believe that it could lead to overestimations of C$_{ANT}$ by the TrOCA method. Based on the suggestions of referee #1, we have rephrased this section to be clearer.**
*Finally, it should also be mentioned that, due to the presence of one of the main OMZ area, denitrification occurs in the eastern South Pacific and can be traced by the N\* parameter (Gruber and Sarmiento, 2007). Denitrification, by transforming organic carbon to inorganic carbon without consumption of oxygen, could induce an overestimation of C$_{ANT}$ by the TrOCA method (and other back calculation methods) due to a biological release of C$_T$ that is not corrected in the formulation of the quasi conservative TrOCA tracer. Horizontal advection by the south equatorial current of the strong negative N\* signal originating from the Eastern Pacific towards the western Pacific was previously described (Yoshikawa et al., 2015). Fumenia et al. (2018) have estimated N\* along the OUTPACE transect and show slightly negative N\* values in the upper thermocline waters at the eastern side of the OUTPACE transect where the highest CANT values are estimated. However, Murata et al. (2007) showed that, based on a direct relationship between CT and N\*, the influence of denitrification should be negligible on C$_{ANT}$ estimations in this area. Therefore, the N\* correction has not been introduced in the C$_{ANT}$ estimates and the effect of denitrication was not quantified here.*

319. The change in the sampling grids means you can't trust these linear fits in the MA region. Your measurements in MA are south of the P21 section where, being closer to the ventilation regions for AAIW and SAMW, you would expect higher Cant. The fact that your measurements are higher relative to P21 here than elsewhere is potentially attributable to that alone.
**As we mentioned earlier, the estimated trends with GLODAP are not only based on the P21 section in the considered area. We agree that by being closer to the ventilation regions for AAIW and SAMW, we would expect higher C$_{ANT}$. However, a careful observation of Fig. 5(c), does not seem to indicate that the trend is due to the latitudinal position of the observations considered. Moreover, the trend is also observed without considering the OUTPACE dataset (see the new table 2).**

325 is -> are
**We agree with this correction.**

Section 7. Here the authors compare their subsurface pHT changes to some surface pHT changes in literature. These are not valid comparisons because subsurface Cant is frequently lower and because the impact of Cant on pH is increased in the surbsurface where Revelle factors are higher.
**We agree that these comparisons are subject to caution. However, the aim of this comparison was to compare the order of magnitude. We propose to add a sentence to point out the differences in buffer factors that exist between surface and subsurface.**
*These rates of acidification are higher than the values reported by Waters et al. (2011) in the Western South Pacific along the P06 Line (south of OUTPACE area at 32°S) between two visits in 1992 and 2008. They are also higher than the surface rates of $pH_T$ decrease of $-0.0016 \pm 0.0001\ a^{-1}$ recorded at the HOT time-series station in the tropical North Pacific and of $-0.0017 \pm 0.0001\ a^{-1}$ and $-0.0018 \pm 0.0001 a^{-1}$ in the tropical North Atlantic at BATS and ESTOC stations respectively (Bates et al., 2014). Differences in buffer factors between surface and subsurface can partially explain these differences. Nevertheless, our results in subsurface ($\sigma_{\theta\,25}$) based on GLODAPv2 and OUTPACE data ($C_T$ and $A_T$), are similar to $pH_T$ trends derived from fCO2 surface observations (e.g. Lauvset et al, 2015).*

328. why 20C?
**20°C represents the mean temperature on the sigma level 25 (20.2 +- 0.7 °C). This has been added in the manuscript.**

376. observations
**We agree with this correction.**

Table 3. Commas are used for decimal points at places in this manuscript while periods are used in other places.
**We agree with the referee that a mixture of commas and periods were used for decimals in the submitted manuscript. We will use consistently periods for decimal points over the entire manuscript.**

Figures. The section plots are tessellated (faint lines going everywhere on them), which is a problem that seems to happen for Matlab 2014b+ when exporting to vector graphics. Consider exporting to high resolution raster files instead. Please ignore if this is just a function of the review-proof.
**We are not able to identify this problem on the section plots we have produced (Using the Generic Mapping Tools, GMT, Version 5.2.1, see [http://gmt.soest.hawaii.edu/](http://gmt.soest.hawaii.edu/)). However, we will of course correct any problems on the figures if the problem persists.**

Colormaps: With the exception of Figure 6 (which would be impossible for people who are red-green colorblind to read), there are no changes to the colormaps that need to be made for this paper to be publishable. However, the authors should give this resource a read:
https://matplotlib.org/cmocean/
At the end of the webpage there are links to papers making the case that rainbow colormaps are not ideal for communicating science. The rest of the page is dedicated to providing alternatives
**We thank the referee for sharing this interesting information on the use of color palettes. After reading some of the links suggested by reviewer, we have changed the color palettes on Figure 6 from red-green to blue-green which seems to be more adapted to colorblind readers. Concerning the rainbow color palette, we discovered with interest all the disadvantages of this none sequential color palette. However, we believe that, in the case of our figures, the rainbow color palette does not lead to a misinterpretation of the data that would justify a change.**

[revised manuscript text omitted]

Figure 1

[Figure]

Figure 6